# Macroscopic photonic single crystals via seeded growth of DNA-coated colloids

Alexander Hensley[1,2], Thomas E. Videbæk [1], Hunter Seyforth [1], William M. Jacobs [2]✉ & W. Benjamin Rogers [1]✉

Photonic crystals—a class of materials whose optical properties derive from their structure in addition to their composition—can be created by self-assembling particles whose sizes are comparable to the wavelengths of visible light. Proof-of-principle studies have shown that DNA can be used to guide the self-assembly of micrometer-sized colloidal particles into fully programmable crystal structures with photonic properties in the visible spectrum. However, the extremely temperature-sensitive kinetics of micrometer-sized DNA-functionalized particles has frustrated attempts to grow large, monodisperse crystals that are required for photonic metamaterial applications. Here we describe a robust two-step protocol for self-assembling single-domain crystals that contain millions of optical-scale DNA-functionalized particles: Monodisperse crystals are initially assembled in monodisperse droplets made by microfluidics, after which they are grown to macroscopic dimensions via seeded diffusion-limited growth. We demonstrate the generality of our approach by assembling different macroscopic single-domain photonic crystals with metamaterial properties, like structural coloration, that depend on the underlying crystal structure. By circumventing the fundamental kinetic traps intrinsic to crystallization of optical-scale DNA-coated colloids, we eliminate a key barrier to engineering photonic devices from DNA-programmed materials.

DNA-programmed self-assembly leverages the chemical specificity of DNA hybridization to stabilize user-prescribed crystal structures[1,2]. Pioneering studies have demonstrated that DNA hybridization can guide the self-assembly of a wide variety of nanoparticle crystal lattices, which can grow to micrometer dimensions and contain millions of particles[3–9]. Attention has now turned toward the goal of assembling photonic crystals from optical-scale particles (i.e., roughly 100–1000 nm in diameter)[10–12] using DNA-programmed interactions. To this end, progress over the past decade has established that DNA can indeed program the self-assembly of bespoke crystalline structures from micrometer-sized colloidal particles[13–19]. However, growing single-domain crystals comprising millions of DNA-functionalized, micrometer-sized colloidal particles remains an unresolved barrier to

the development of practical technologies based on DNA-programmed assembly. Prior efforts have yielded either single-domain crystals no more than a few dozen micrometers in size[13–16] or larger polycrystalline materials with heterogeneous domain sizes[12,15,17,20]. These features—small crystal domains, polycrystallinity, and size dispersity—have therefore precluded the use of DNA-coated colloidal crystals in photonic metamaterial applications.

Assembling macroscopic materials from DNA-functionalized, micrometer-sized colloids is challenging due to the vastly different length scales between the DNA molecules and the colloidal particles (Fig. 1a). This combination leads to crystallization kinetics that are extremely sensitive to temperature and prone to kinetic trapping[1,21–23]. The resulting challenges are both practical and fundamental in nature.

[1]Martin A. Fisher School of Physics, Brandeis University, Waltham, MA 02453, USA. [2]Department of Chemistry, Princeton University, Princeton, NJ 08544, USA. ✉e-mail: wjacobs@princeton.edu; wrogers@brandeis.edu

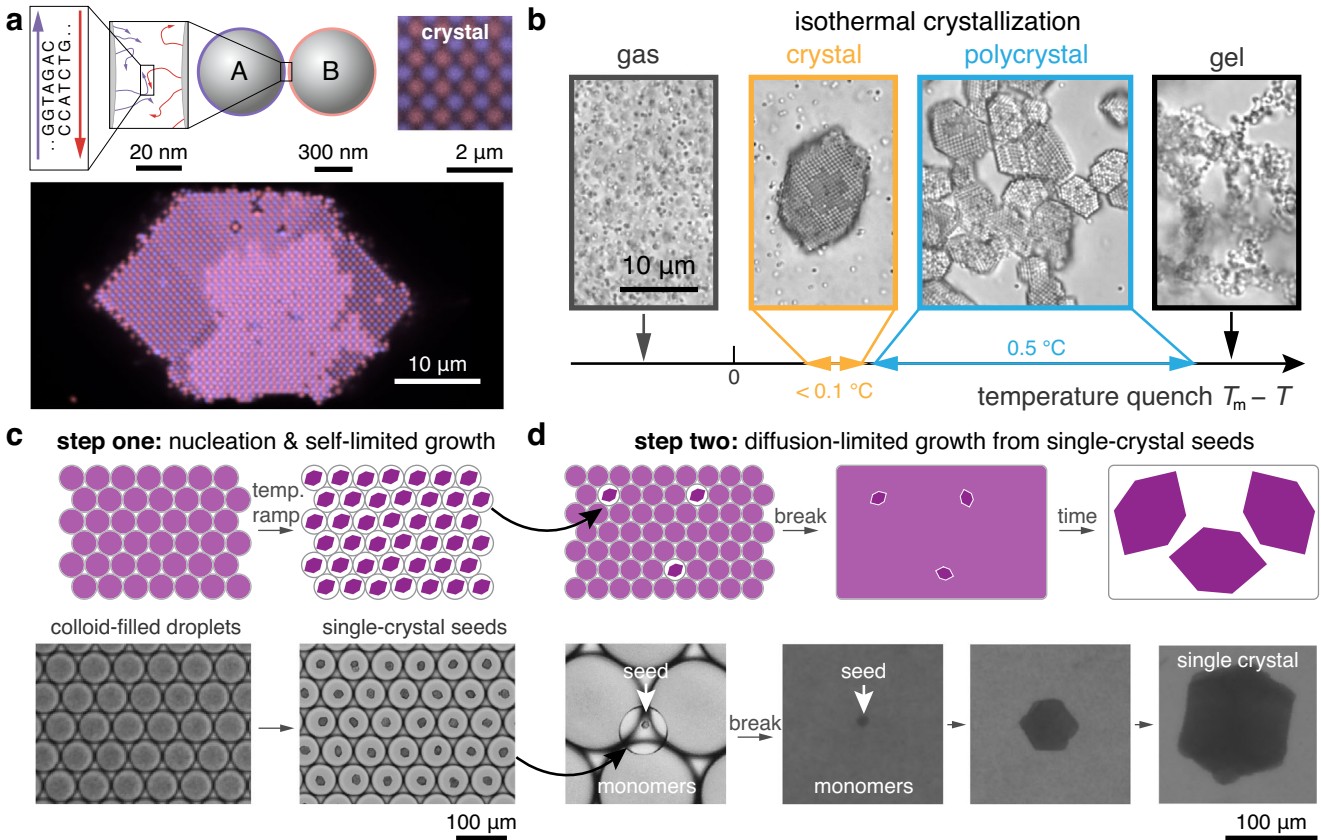

**Fig. 1 | Large single-domain crystals can be self-assembled from DNA-coated colloids via a two-step process. a** DNA molecules at the nanometer scale can link together micrometer-scale colloidal particles to program the assembly of colloidal crystals. The fluorescence micrograph shows a DNA-programmed binary colloidal crystal formed from 600-nm-diameter particles. The inset micrograph shows the crystal lattice. Blue and red correspond to particle species A and B, respectively. **b** DNA-programmed crystallization is strongly temperature dependent. Below the melting temperature, $T_m$, single crystals can form in a very narrow temperature window, indicated in orange. Just below this temperature window, kinetically arrested polycrystalline (blue) or gel-like assemblies form (black). Optical micrographs show examples of these different states for the same particles as in (**a**). **c** The first step of our protocol involves nucleating size-monodisperse single crystals in monodisperse water droplets made via microfluidics, shown in a cartoon schematic (top) and in brightfield micrographs (bottom). Monodisperse droplets filled with DNA-coated colloids are slowly cooled to produce same-size single crystals. **d** The second step involves recovering the single crystals by breaking the emulsion and then using them to seed crystal growth in a metastable colloidal suspension, shown in a schematic (top) and in brightfield micrographs (bottom). A small number of crystal-containing droplets from (**a**) are combined with droplets containing DNA-coated particles. The emulsion is ruptured, and the system is cooled to a temperature at which crystals grow, but the nucleation of additional crystals is suppressed.

For example, recent work has shown that crystal nucleation rates can vary by orders of magnitude over a temperature range of only 0.25 °C[19]. Extremely precise temperature control would therefore be required to self-assemble single-domain crystals from a bulk solution (Fig. 1b). At the same time, annealing polycrystalline materials is difficult due to the combination of the short-range attraction and the friction arising from the DNA-mediated colloidal interactions, which slows the rolling and sliding of colloidal particles at crystalline interfaces[15,19,24,25]. Thus, the impracticality of assembling and annealing macroscopic crystals composed of DNA-functionalized, micrometer-sized particles results from intrinsic features of these materials that cannot be easily designed around.

In this article, we describe a two-step approach to overcome these limitations and self-assemble macroscopic photonic crystals from micrometer-scale DNA-coated colloidal particles. We first show that small, monodisperse single crystals can be reliably assembled within nanoliter-scale microfluidic droplets by subjecting the droplets to a simple cooling protocol (Fig. 1c). Then, to go beyond a fundamental size limitation imposed by the droplet-confined assembly, we use these crystals to seed continued growth in a bulk solution, while simultaneously suppressing further crystal nucleation (Fig. 1d). We develop a theoretical framework that models both processes quantitatively, enabling us to make monodisperse suspensions of single

crystals with prescribed dimensions and predictable yields. Finally, by varying the size of the colloidal particles and the duration of the secondary growth phase, we show that our approach can be easily generalized to create a variety of monodisperse macroscopic crystals with different crystal structures and, therefore tunable photonic properties, including crystals that can be seen by the naked eye. Most importantly, we emphasize that this approach for synthesizing macroscopic crystals is robust, meaning that the procedure is insensitive to small changes in processing conditions, can be repeated over and over again, and can be applied to different particle sizes and compositions. Therefore, our platform could enable significant future advances in DNA-programmed assembly of both nanometer- and micrometer-scale colloids.

## Results
### Assembly of monodisperse crystalline seeds
We first seek to understand the physics governing the self-assembly of colloids confined to small water droplets subject to a time-dependent cooling protocol (step one in Fig. 1c). Whereas we previously demonstrated that small single-domain colloidal crystals could be assembled in monodisperse water droplets[19], here we aim to understand how to optimize the yield of single-domain crystals assembled via this approach. We thus perform systematic experiments for a particular

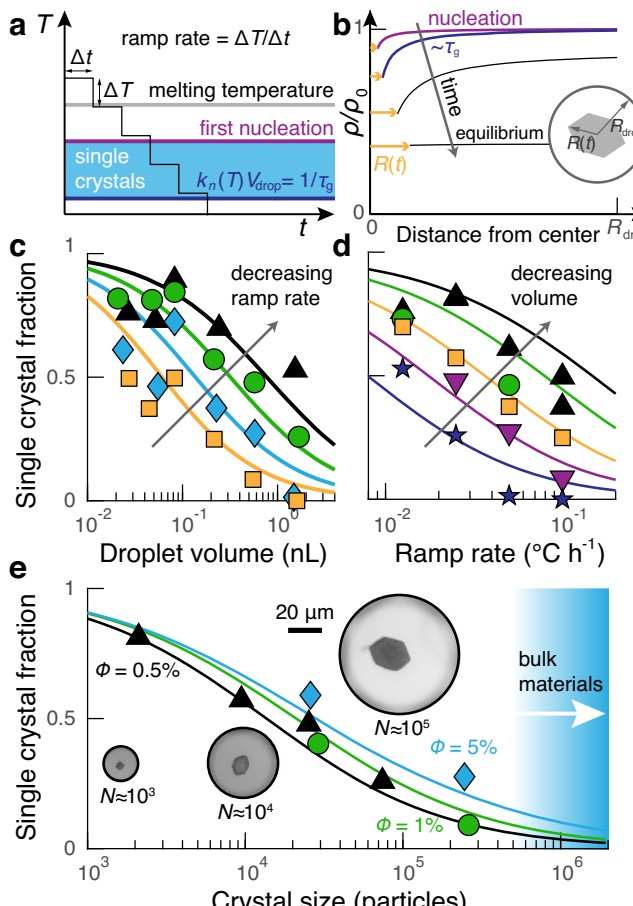

**Fig. 2 | Linear cooling produces single crystals within nanoliter droplets. a** The temperature ramp is a series of discrete 0.1 °C steps with a duration of $\Delta t$, which starts above the melting temperature and runs until all of the particles are incorporated into the crystals. **b** The relative colloid concentration as a function of the distance from the center of the crystal at different times during crystal growth. Curves show the predicted concentration profiles, and orange arrows indicate the crystal radius, $R(t)$; $\rho$ is the colloid concentration, and $\rho_0$ is the uniform concentration at time $t = 0$. **c** The fraction of droplets that form single crystals as a function of droplet volume. Points represent experiments, and lines show model predictions for different ramp rates: 0.0125 °C h⁻¹ (triangles), 0.025 °C h⁻¹ (circles), 0.05 °C h⁻¹ (diamonds), and 0.1 °C h⁻¹ (squares). **d** The data in **c** replotted versus the ramp rate. Different shapes represent different droplet volumes: 0.04 µL (upward triangles), 0.08 µL (circles), 0.2 µL (squares), 0.5 µL (downward triangles ), and 1.4 µL (stars). **c**, **d** Share the same y-axis. **e** The fraction of single crystals versus the number of particles within each droplet using a ramp rate of 0.025 °C h⁻¹. Different symbols correspond to different particle volume fractions. The single-crystal fraction decreases monotonically with decreasing particle concentration and increasing crystal size. In all cases, the data and model predictions approach zero before crystals reach sizes of $10^6$ particles, which we define as the threshold for macroscopic materials. The particles are 600 nm in diameter in (**c**–**e**), and each data point corresponds to a distinct sample.

prototypical crystal structure, nearly isostructural to copper gold (CuAu)[26] (see Suppl. Note 3), to validate a parameter-free analytical theory, which provides insights into the fundamental physical limitations of this method, as we discuss below.

Using microfluidics, we create a water-in-oil emulsion of nanoliter-scale droplets that are filled with a binary suspension of 600-nm-diameter DNA-coated polystyrene particles and subject the droplets to a linear cooling protocol. Throughout the protocol, the temperature is decreased in a staircase fashion with a specified time delay $\Delta t$ and a temperature change $\Delta T = -0.1$ °C (Fig. 2a). Because the DNA-programmed attractions become stronger with decreasing

temperature, one or more crystals eventually nucleate and grow within each droplet (Fig. 2b), leading to an ensemble of small crystals throughout the emulsion. The temperature ramp then continues until all the colloidal particles in each droplet are incorporated into the crystal phase. At the conclusion of the temperature ramp, we measure the fraction of droplets within the ensemble that contain precisely one single-domain crystal (see Suppl. Fig. 2 for an example). Importantly, because the final crystal phase incorporates all the colloidal particles, single-domain crystals assembled via this technique are monodisperse in volume to within the typical 5% variation in the initial particle loading[19].

We find that the yield of single crystals decreases with increasing droplet volume and increasing ramp rate. Both of these trends can be rationalized by considering the pathway by which single crystals form: The growth of the first crystal that nucleates must reduce the supersaturation throughout the droplet in order to suppress the nucleation of additional crystals (Fig. 2b). We, therefore, consider the factors that influence both the temperature at which the first crystal is most likely to nucleate and the probability of subsequent nucleation events. We propose that varying the droplet size (Fig. 2c) primarily affects the single-crystal fraction by altering the probability of secondary nucleation events. Because the droplet size has a relatively small influence on the initial nucleation temperature, the likelihood of secondary nucleation is determined by the time required to reduce the supersaturation by particle diffusion. Therefore, at a fixed ramp rate, the single-crystal fraction is smaller in larger droplets because the diffusive time scale is proportional to the square of the droplet diameter. By contrast, we propose that changing the ramp rate (Fig. 2d) primarily affects the initial nucleation temperature at which the first crystal is formed within the droplet. Specifically, at a fixed droplet volume, faster ramp rates strongly bias the initial nucleation event towards lower temperatures as a direct result of the reduced duration of the higher-temperature steps. Because particle diffusion is comparatively insensitive to temperature, the dominant effect of a faster ramp rate is likely to be the increased instantaneous nucleation rate at the lower first-nucleation temperature, which in turn increases the probability of additional nucleation events.

To describe these factors quantitatively, we introduce the growth time, $\tau_g$, to describe the typical time required for crystal growth to suppress further nucleation elsewhere in the droplet. Intuitively, the nucleation rate should be slower than $1/\tau_g$ in order for a single crystal to assemble (Fig. 2a). For example, in an isothermal protocol, the probability that a second crystal does not nucleate is given by $\exp[-V_{drop}k_n(\rho_0, T)\tau_g]$, where $V_{drop}$ is the droplet volume and $k_n$ is the nucleation rate density at the initial colloid concentration $\rho_0$ and temperature $T$. Even though the situation is more complicated in the case of a temperature ramp, $\tau_g$ is conceptually and computationally useful because it is only weakly dependent on temperature (see Suppl. Note 4A, B for a formal definition and further details).

Theoretical predictions based on a modified classical theory of nucleation and growth enable us to predict $\tau_g$ and thus to describe our experimental results quantitatively with no adjustable parameters. Using previously determined concentration and temperature dependencies of the nucleation rate and crystal growth velocity[19], we first predict the probability of the initial nucleation event and the ensuing crystal growth dynamics for a prescribed droplet size and temperature protocol. Importantly, this model captures the evolution of the concentration field around the first crystal that nucleates, allowing us to predict the decrease in the supersaturation throughout the entire droplet volume as this initial crystal grows, which is necessary to compute $\tau_g$ quantitatively (Fig. 2b and Suppl. Fig. 13). We then use these calculations to predict the probability of growing a single crystal in a droplet during a time-dependent temperature protocol using a generalization of the isothermal expression given above (Suppl. Fig. 14; see Suppl. Note 4C for details).

Across nearly the full range of parameter space, we find that our model captures our measured single-crystal fractions quantitatively, within the uncertainty of our experimental measurements (Fig. 2c, d). We note that systematic deviations between the predictions and measured single-crystal fraction are observed at the slowest ramp rates, for which the measurements exhibit nonmonotonic dependencies on the ramp rates and droplet volumes. However, we attribute these effects to substantial evaporation of the solvent from the droplets, which can reduce the smallest droplet volumes by as much as 30% during annealing. Therefore, the overall accuracy of our model allows us to predict the conditions required to achieve a target yield and to optimize the temperature-ramp protocol subject to a maximum duration of the experiment.

Unexpectedly, our theoretical model predicts that utilizing discrete temperature steps, as opposed to a continuous ramp, is, in fact, beneficial for maximizing the single-crystal yield at a fixed ramp rate, $|\Delta T/\Delta t|$ (Suppl. Fig. 15; see Suppl. Note 4D for details). As long as $\Delta t$ is longer than $\tau_g$, each discrete step can be considered as an isothermal protocol, which optimally suppresses further nucleation by holding the nucleation rate density, $k_n$, constant for the entirety of $\tau_g$. By contrast, a continuous ramp implies that $k_n$ increases continuously following the first nucleation event, increasing the probability of secondary nucleation events. Nonetheless, $\Delta t$ cannot be made too large, as the correspondingly large temperature steps required to maintain the fixed ramp rate would tend to bias the first nucleation event to lower temperatures, and thus nucleation rates that are faster than $1/\tau_g$. Balancing these competing factors, our model predicts that the single-crystal probability is maximized for temperature steps on the order of $\Delta T = 0.1\,°C$. As this is the step size used in our experiments, our modeling suggests that further refinement of the precise functional form of our temperature protocol would yield minimal improvement (Suppl. Fig. 16). In practice, we, therefore, only need to tune the step duration, $\Delta t$, to achieve a target single-crystal yield using a prescribed droplet volume and particle concentration.

Despite its many advantages, the temperature-ramp protocol is not a magic bullet for assembling macroscopic photonic crystals from DNA-coated colloids since both theory and experiment point to fundamental physical limitations on the size of single crystals that can be assembled in droplets at high yield. Since larger droplet volumes reduce the yield, growing larger crystals at a fixed ramp rate can only be achieved by increasing the particle concentration. Yet, when comparing different cooling protocols that would yield a given final crystal size, we find that increasing the initial colloidal concentration to as much as 5% by volume has only a modest effect on the single-crystal yield (Fig. 2e). On the other hand, we encounter a practical limitation when reducing the ramp rate below $0.025\,°C\,h^{-1}$, since extremely long annealing protocols are accompanied by substantial evaporation of the solvent from the droplets as noted above. Thus, taken together, our model and experiments demonstrate that single-domain crystals containing more than one million DNA-functionalized, micrometer-sized particles cannot be assembled in droplets within a practical annealing duration with any appreciable yield, regardless of the precise form of the temperature protocol.

**Diffusion-limited seeded growth**

To overcome this fundamental limitation, we introduce a second processing step in which the small, monodisperse single crystals assembled in droplets can be grown by orders of magnitude in size (step two in Fig. 1d). This second step exploits the fact that particles are able to attach to a crystal surface at a higher temperature, and thus lower supersaturation, than that at which nucleation occurs. To this end, we transfer the droplet-assembled crystals to a fresh set of droplets containing 'weak' particles, whose DNA grafting density is reduced by half during particle synthesis, and rupture the emulsion. The resulting lower melting temperature of the weak particles allows

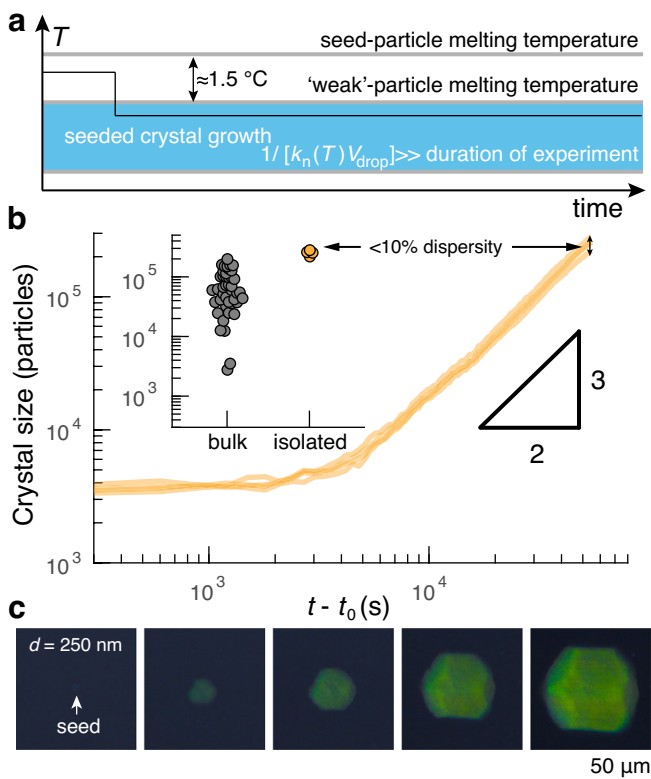

**Fig. 3 | Seeded crystal growth is diffusion-limited and yields monodisperse large crystals. a** The temperature protocol for the seeded growth step. The supercooling is kept roughly constant during the experiment such that the seeds can grow, but homogeneous nucleation is suppressed. **b** A plot of the crystal size of four seeds in the same sample over time. The particles are 600 nm in diameter. We define $t_0$ to be the time at which the crystal has grown by ten percent in size. In all cases, the growth is diffusion limited with a characteristic 3/2 power scaling. In this specific experiment, the crystals are spaced apart by roughly four times their final diameter. The inset shows the crystal sizes at the end of the experiment for seeded growth (orange, $n = 4$) and bulk nucleation and growth (gray, $n = 43$). **c** Micrographs of seeded growth of 250-nm-diameter particles imaged in transmission through crossed polarizers show the appearance of coloration upon the growth of a single crystal.

them to exist in the gas phase at temperatures below the melting temperature of the seed crystals themselves. We then anneal the system at a temperature above the homogeneous nucleation temperature of the weak particles but below the melting temperature of the crystalline seed (Fig. 3a and Suppl. Fig. 1). Because the seed crystal size itself is not essential, we choose seeds from a slow ramp protocol that uses a ramp rate of $0.025\,°C\,h^{-1}$ and a droplet volume of roughly 0.05 nL, resulting in a single-crystal fraction of nearly 100%.

We track the size of the single-crystalline seeds during secondary growth to infer their growth mechanism. More specifically, we measure the crystal area as a function of time and then relate the measured two-dimensional area to the number of particles per crystal, $N(t)$, using an empirical calibration curve (Fig. 3b). Since we expect that seeds will be effectively isolated from one another if their diffusion fields do not strongly overlap, we dilute the seeds until they are separated by at least a few multiples of the target crystal diameter at the end of the secondary growth experiment. Consistent with this expectation, growth curves of isolated seeds confirm that seeded crystals can be grown by orders of magnitude in size and that their size follows the scaling relation predicted by diffusion-limited growth, with $N(t) \propto t^{3/2}$ at long times (Fig. 3b and Suppl. Fig. 17). Note that at early times, the crystal size does not follow the same power-law scaling because the crystals begin with a finite initial size of roughly 3000 particles per crystal in

this experiment (see Suppl. Note 2B, C for details on the experiment and calibration).

Importantly, because diffusion-limited growth is deterministic, the size distribution of the final crystals after the second growth step retains the monodispersity of the original crystal seeds. Figure 3b shows that the standard deviation of the final crystal volumes is less than 10% of the mean. By contrast, the final sizes of colloidal crystals nucleated in a bulk solution vary by roughly two orders of magnitude, reflecting both the distribution of nucleation times as well as competition for free particles from neighboring growing crystals (Fig. 3b, inset, and Suppl. Fig. 18). Thus the deterministic growth behavior in our approach not only allows us to predict the duration of the secondary growth phase required to grow a single-domain crystal of a prescribed size, but it also allows us to make many same-sized macroscopic colloidal crystals in a single experiment. In principle, there is no upper limit to the size of crystals that can be prepared using our approach; growing even larger crystals simply requires reducing the seed density and increasing the growth time.

To demonstrate the direct connection between the bulk optical properties and the crystal size, we perform the same seeded-growth experiment using particles that are comparable to half the wavelength of visible light. Figure 3c shows snapshots from a typical growth trajectory for particles that are 250 nm in diameter. Whereas the seed is almost imperceptible, we see that the crystal exhibits prominent coloration upon growth and that its color saturation increases until the crystal is roughly 50 micrometers in linear dimension (i.e., containing a few million particles.) This simple experiment highlights two critical features of the second seeded-growth step: First, the final snapshot shows that seeded growth can create crystals containing millions of particles that cannot be directly assembled in droplets; and second, achieving crystals of this size scale is essential to realizing the bulk metamaterial properties of single-crystalline assemblies of colloids.

## Macroscopic photonic crystals

Finally, we turn our attention toward using our two-step platform to assemble macroscopic single-crystalline materials from DNA-coated colloids. To this end, we first confirm that the seeded crystals are indeed single crystalline by imaging them at high magnification with single-particle resolution. An example of a crystal assembled from 600-nm-diameter particles is shown in Fig. 4a. To demonstrate that the crystal was indeed grown from a seed, we use undyed seed particles and fluorescently labeled 'weak' particles. The seed is clearly visible in the interior of the crystal, as is the seed outline or the seed crystal habit (Fig. 4a). Zooming in on the crystal lattice, we see the crystalline order with single-particle resolution. Furthermore, we see that a scaled version of the seed outline follows the crystallographic directions of the lattice, indicating that growth preserves the crystal structure of the seed. Similarly, we see that a scaled version of the seed outline also approximates the habit of the large-scale seeded crystal itself. Therefore, we conclude that the assembled structure—which contains roughly one million particles—is a single crystal.

To further drive home the importance of our two-step approach and place it within the broader context of DNA-programmed crystallization, we plot the estimated crystal sizes from the literature versus the particle diameter. Figure 4b illustrates the challenge that the field of DNA-programmed assembly has faced in making bulk self-assembled materials from optical-scale particles: Whereas the largest single crystals made from 10-nm-diameter DNA-coated colloids contain approximately $10^{10}$ particles[9], the inherent kinetic challenges associated with using roughly 10-nm-long DNA to direct the crystallization of micrometer-scale particles[27] have limited such crystals to $10^5$-fold fewer particles per crystallite[19,20]. A direct consequence of this hurdle is that, prior to our work, the region of parameter space corresponding to bulk photonic crystals was completely vacant (Fig. 4b,

orange box). See Suppl. Note 2B for a description of our method of estimating these crystal sizes.

Our two-step approach enables us to populate this space and create macroscopic photonic crystals using DNA-programmed self-assembly. We demonstrate the potential of our experimental platform by executing our two-step method to make crystals that exhibit bulk optical properties using particles with diameters of 600 nm (Fig. 4c), 430 nm (Fig. 4d), and 250 nm (Fig. 4e). For each particle size, we successfully self-assemble similarly large single crystals that rival the sizes of crystals formed from nanometer-scale particles (see Fig. 4b for comparison). Furthermore, we find that each of these particle sizes assembles into a different crystal structure, indicating that our method can be applied to the synthesis of different crystal symmetries. The 600-nm-diameter particles assemble into a crystal that is nearly isostructural to CuAu[26]; the 250-nm-diameter particles assemble into a crystal that is isostructural to CsCl; and the 430-nm-diameter particles assemble into a body-centered tetragonal (BCT) crystal structure that is intermediate between the CsCl and CuAu. We hypothesize that these crystal structures form due to a balance of specific attraction between 'unlike' particles due to DNA hybridization and nonspecific attraction between both 'unlike' and 'like' particles due to van der Waals forces[23]. Because the precise crystal structures are not the primary focus of this article, we have placed a detailed discussion of their characterization in Suppl. Note 3.

Owing to the size of the constituent building blocks, together with the size and quality of the crystalline assemblies, our crystals show pronounced photonic properties. For example, the 250-nm-diameter-particle crystals exhibit a prominent stop band for frequencies corresponding to red light and therefore exhibit structural coloration in reflected light (Fig. 4e). The 400-nm-diameter-particle crystals also exhibit coloration in reflection (Fig. 4d). We hypothesize that the green-yellow structural color of the 400-nm-diameter-particle crystals arises from second-order diffraction, which explains why the apparent color is shorter wavelength even though the particle size is larger than that of the 250-nm-diameter particles (see Suppl. Note 3F for details).

We stress that the specific photonic properties of our crystals are not the most exciting result, as large colloidal crystals exhibiting structural coloration have been made by numerous other methods[28–32]. Rather, the notable achievement here is that our method enables the robust, near-to-equilibrium assembly of macroscopic single crystals from DNA-coated micrometer-scale colloids that can grow to macroscopic sizes visible to the naked eye (Fig. 4b and Suppl. Fig. 19). The importance of demonstrating the assembly of macroscopic crystal domains from DNA-coated micrometer-scale colloids is that many of the other reported methods for making large-scale colloidal crystals lack the programmability of DNA, which is essential for creating complex crystalline lattices, like cubic diamond[12], for advanced photonic bandgap materials. Furthermore, we expect that our two-step platform is applicable to any Brownian suspension of DNA-coated colloids (i.e., particles with diameters less than roughly two micrometers) and, because the interactions that drive nucleation and growth are due entirely to DNA hybridization, should also apply to a wide range of colloidal particle compositions, including polymers, metals, metal oxides, semiconductors, and magnetic materials[4,5,15,33]. We similarly anticipate that our theoretical model of droplet-confined nucleation and growth under a time-dependent protocol could be extended to other particle sizes and crystal structures with minimal modifications.

## Discussion

In summary, we have developed a platform to create macroscopic single crystals from DNA-coated colloids by decoupling nucleation and growth. Our approach solves a number of longstanding challenges associated with isothermal nucleation in bulk solution and the resulting heterogeneous distribution of relatively small self-assembled

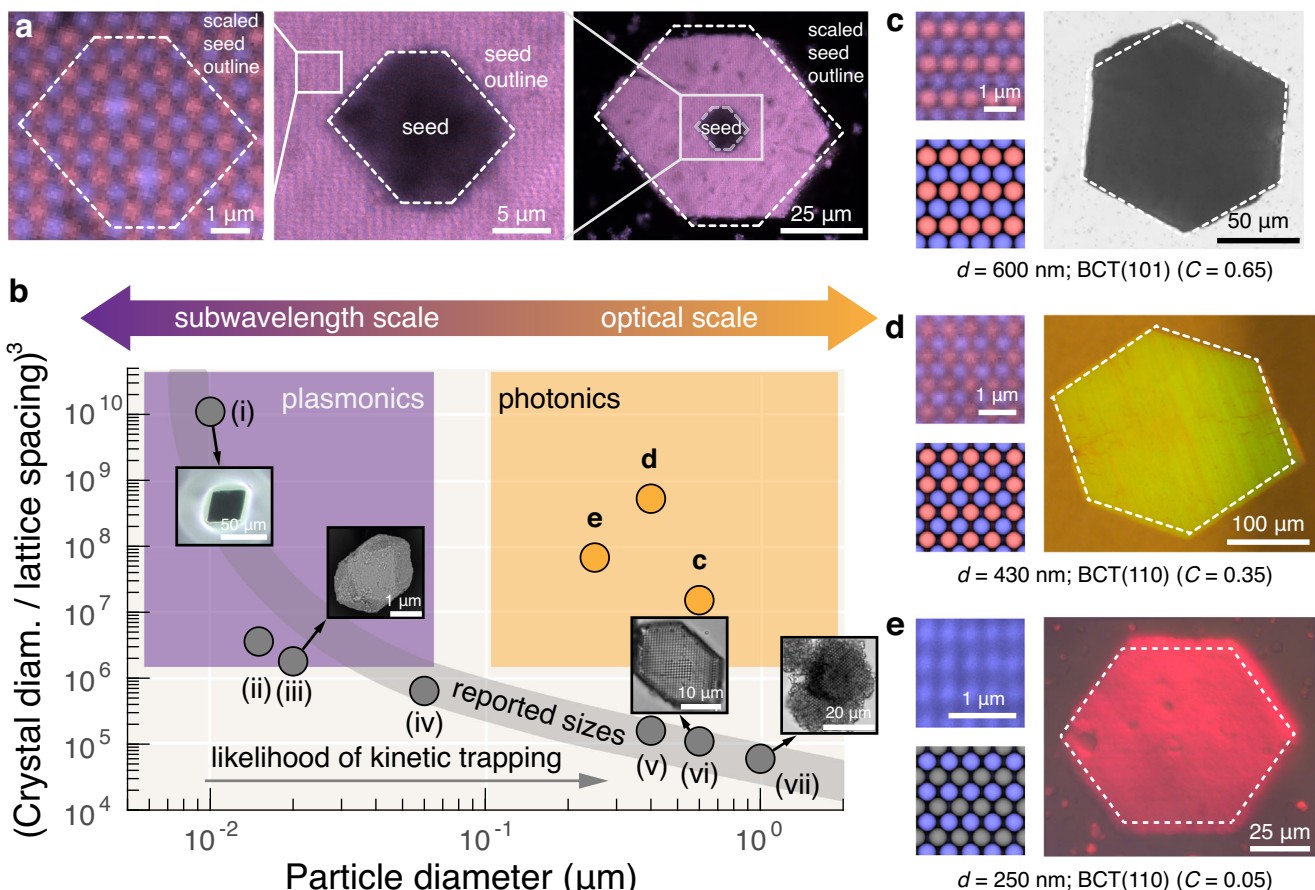

**Fig. 4 | Single crystals from optical-scale particles can grow to macroscopic dimensions and exhibit photonic properties dependent on the crystal structure. a** Confocal fluorescence images of a seeded crystal of 600-nm-diameter particles. The crystal has a well-defined habit (right) that is consistent with the seed habit (middle) and the underlying lattice structure of the crystal (left). The seed particles are not dyed. **b** An overview of the reported sizes of the largest crystals of DNA-coated particles from the literature as a function of the constituent particle size, spanning subwavelength- to wavelength-scale particles. Generally, the number of particles per single-domain crystal decreases as the particle size increases due to kinetic trapping (gray points). Our two-step protocol breaks this trend, allowing well-faceted crystals of optical-scale DNA-functionalized particles to be grown multiple orders of magnitude larger than before. Orange points show the single-crystal sizes for three particle diameters, shown in (**c**–**e**). Each point is from a distinct sample. **c** A brightfield micrograph of a single-crystal of 600-nm-diameter particles, which has a crystal habit (dashed outline) consistent with the (101) view of a body-centered tetragonal (BCT) crystal structure (BCT parameter $C = 0.65$) as shown in the insets: experiment (top) and model (bottom). **d** A micrograph imaged in reflection through crossed polarizers of a single-crystal of 430-nm-diameter particles, which has a crystal habit (dashed outline) consistent with the (110) view of BCT ($C = 0.35$), as shown in the insets: experiment (top) and model (bottom). **e** A micrograph imaged in reflection through crossed polarizers of a single-crystal of 250-nm-diameter particles, which has a crystal habit (dashed outline) consistent with the (110) view of BCT ($C = 0.05$), as shown in the insets: experiment (top) and model (bottom); we only show one particle species because the other species' dye emits in the red so the particles are below the diffraction limit and cannot be resolved. The crystal structure in (**c**) is closest to CuAu; the structure in (**d**) is intermediate between CuAu and CsCl; and the structure in (**e**) is isostructural to CsCl. All crystals were grown for roughly two days. Literature points and corresponding micrographs are reproduced from the following references: (i) adapted with permission from[9] (copyright 2022 Springer Nature), (ii)[8], (iii) adapted with permission from[6] (copyright 2022 Springer Nature), (iv)[42], (v)[19], (vi) adapted with permission from[19] (copyright 2022 National Academy of Sciences), and (vii) adapted with permission from[20] (copyright 2020 American Chemical Society).

crystals. We first showed that our method for assembling monodisperse crystalline seeds in droplets is theoretically near-optimal, despite fundamental physical constraints on the size of seeds that can be formed in this manner. We then demonstrated a practical and reliable strategy for controlling secondary growth in a bulk solution, which preserves the narrow size distribution due to the deterministic nature of diffusion-limited growth. Although seeded growth of colloidal crystals has been explored before, previous attempts have focused primarily on growing thin shells[34], growing crystals from two-dimensional templates[35–37], or using seeds that do not match the thermodynamically favored crystal structure[38]. By contrast, our method uses three-dimensional seeds that are isostructural to the crystal that we wish to grow. The end result is a monodisperse distribution of single crystals whose sizes are precisely controlled by the duration of the seeded growth process.

By growing single crystals more than two orders of magnitude larger than was previously possible using optical-scale DNA-coated colloids, our protocol accomplishes the longstanding goal of assembling DNA-programmed materials with user-specified photonic properties. We fully expect that our method could be extended to make single crystals with other crystallographic symmetries[4,39], including ones that can only be synthesized by DNA-programmed assembly, such as colloidal diamond[12], by changing the constituent particles. Furthermore, incorporating additional strategies for processing same-size single-crystalline assemblies, as was recently demonstrated using nanoscale building blocks[40], could open up additional possibilities for hierarchical materials engineering. By providing robust routes to assembling bespoke metamaterials, the advances developed herein promise to bring DNA-programmed colloids out of the lab and into practical use.

## Methods

### Synthesizing DNA-coated colloids

Colloidal particles are functionalized with DNA using a combination of strain-promoted click chemistry and physical grafting, following a modified version of the methods described by Pine and co-workers[41]. We first functionalize PS-b-PEO ($M_w = 6500$ g mol$^{-1}$ PEO and 3800 g mol$^{-1}$ PS, Polymer Source, Inc.) with an azide group. More specifically, we attach a mesyl group via methanesulfonyl chloride (471259, Sigma-Aldrich), and then we replace the mesyl group with an azide group N$_3$ via sodium azide (S2002-5G, Sigma-Aldrich). Next, we incorporate the PS-b-PEO-N$_3$ onto the surface of 600-nm-diameter polystrene particles (S37495, Molecular Probes) by swelling the particles with tetrahydrofuran (THF, 99.9% inhibitor-free 401757, Sigma-Aldrich) in an aqueous solution of PS-b-PEO-N$_3$. Next, we add additional deionized (DI) water to the solution and wait for an hour to dwell on the particles. The particles are then washed five times by repeated centrifugation and resuspension in DI water. Finally, we attach DBCO-modified DNA molecules via a click reaction with the azide groups in a solution of tris-EDTA buffer (pH 8), pluronic F-127 (51181981, Sigma-Aldrich), and sodium chloride. We rotate the reaction mixture end-over-end in an oven at 60 °C for 24 h and then wash the particles five times in DI water. We store the particles at a concentration of 1% (v/v) in DI water at 4 °C.

We study the crystallization of a binary mixture of same-sized DNA-coated colloids. One particle species is coated with sequence A: 5′-(T)51-GAGTTGCGGTAGAC-3′; the other particle species is coated with sequence B: 5′-(T)51-AATGCCTGTCTACC-3′. The sequences are purchased from Integrated DNA Technologies (IDT). All crystallization experiments are performed in 1xTE with 500 mM NaCl. In our experiments, these sequences yield a melting temperature of roughly 52 °C for the 600-nm-diameter particles.

### Fabricating the microfluidic device

Microfluidic drop-makers are fabricated via standard photolithographic techniques. A glob of SU8 (SU-8 2075, or SU-8 3010 Micro-Chem) roughly the size of a quarter is poured onto a silicon wafer (3-76-024-V-B, Silicon Materials Inc.). The wafer is then spun at 500 rpm with a spin coater at a ramp rate of 100 rpm s$^{-1}$ for 5 s and then to between 1000 rpm and 3000 rpm at a ramp rate of 300 rpm s$^{-1}$ for 60 s, the specifics of which will lead to a device thickness between 20 and 80 μm. Next, the wafer is placed onto a 65 °C hot plate for 3 min and then a 95 °C hot plate for 5 min. A photomask (Output City) with the pattern of our microfluidic device is placed on top of the wafer, which is then moved to a Manual Mask Aligner System (ABM-USA) and exposed to UV light for 46 s for a total of 160 mJ. The mask is removed, and the wafer is washed with isopropanol and propylene glycol methyl ether to remove the undeveloped photoresist. The wafer is then dried with an airbrush and placed on a 65 °C hot plate for 3 min and a 95 °C hot plate for 20 min. Next, the wafer is placed in a glass Petri dish with PGME and shaken back and forth for 10 min to remove any photoresist. Finally, the wafer is sprayed with isopropanol and dried with an airbrush.

The master is a negative of the actual device and acts as a mold. Thirty grams of polydimethylsiloxane (PDMS) and 3 g crosslinker (1673921, Dow Chemical Company) are mixed using a Thinky AR-250 planetary centrifugal mixer for 3 min. A plastic Petri dish is lined with aluminum foil, and the microfluidic-device master is placed face-up in the dish. The mixed PDMS is then poured onto the master and placed in a vacuum desiccator for 30 min to remove any bubbles from the PDMS mixture. The dish is placed in a 70 °C oven overnight. The wafer is removed from the dish, the foil is peeled off, and a hobby knife is used to cut away the excess PDMS and separate it fully from the master. A coring tool (69039-07, Electron Microscopy Sciences) is then used to punch holes into all the device inlets and outlets. A glass slide (2947-75X50, Corning) and the PDMS chip are placed into an oxygen

plasma cleaner (Zepto, Diener electronic) for 45 s. The PDMS chip is then laid down onto the glass slide and held with uniform pressure for 30 s, permanently bonding them together.

### Droplet making

Syringe pumps are used to operate the microfluidic device to produce monodisperse droplets containing a colloidal suspension. The channels of the microfluidic device are made hydrophobic by flushing them with Aquapel (B004NFW5EC, Amazon), leaving them for 30 s, and then flushing them again with air to remove the Aquapel. The channels are then flushed with HFE-7500 oil (3M) and air again. Flow rates are controlled independently by three syringe pumps (98-2662, Harvard Apparatus) connected to the device via tubing (06417-11, Cole-Palmer) that is slightly larger in diameter than the holes to ensure a snug fit. HFE-7500 with 2.5% RAN fluorosurfactant (008-FluoroSurfactant-5wtH-20G, RAN Biotechnologies) is fed into the oil inlet, 1 M NaCl in 1xTE buffer is fed into one aqueous inlet, and DNA-coated particles suspended at twice the desired volume fraction in 1xTE are fed into the other aqueous inlet. The particles are created in small quantities, so we cannot load them directly into the syringe. Instead, they are loaded into the tube by using a reverse flow rate and never enter the syringe body. A couple of centimeters of air are left on either side of the particle solution to ensure that the suspension does not mix with the carrier fluid due to Taylor dispersion. The flow rates of the oil and aqueous phases depend on the desired droplet size and the thickness of the microfluidic device being used and are generally between 400 μL h$^{-1}$ and 1000 μL h$^{-1}$. The droplets are deposited from the outlet tube directly into a 0.2 ml Eppendorf tube. As much as 10 μL of HFE-7500 with 2.5% RAN is added to the tube if the ratio of oil to aqueous flow rates is lower than 1:1. A very small amount of droplets are loaded directly into a glass capillary and the droplet size is verified via brightfield microscopy.

### Droplet temperature-ramp experiments

Eppendorf tubes with particle-filled droplets are placed in the central wells of a single module C1000 Touch Thermo Cycler (Bio-Rad). An Eppendorf with a thermistor and thermal paste is placed in a well next to the sample to log the sample temperature. A ramp protocol is used that comprises 30 min of melting at 56 °C followed by a drop in the temperature at which the ramping protocol begins. The ramping protocol involves dropping the temperature in 0.1 °C increments and holding for a specific interval defined by the quoted ramp rate of the experiment. For instance, for a ramp rate of 0.025 °C h$^{-1}$, a 0.1 °C drop every 4 h would be used. The ramp continues for 40 steps covering a range of 4 °C. The starting temperature is decided by placing a small number of particles in a buffer in an Eppendorf and observing whether the particles aggregate and sink over the course of 30 min. Once this transition temperature is found, the starting temperature is set to 1.5 °C above it.

### Fabricating sample chambers

Sample chambers to observe the presence of crystals in the droplets are comprised of a rectangular capillary filled with the microfluidic emulsion that is glued to a glass coverslip. A 200-μm tall, 2-mm-wide glass rectangular capillary (5012, VitroCom) is cut to 3 cm in length with a glass scoring pen and held suspended in place with a pair of clamping tweezers. Approximately 2–3 μL of the droplet emulsion is transferred into the capillary via a micropipette that has been snipped at the tip to have a wider inlet. HFE 7500 with 2.5% RAN is used to fill the rest of the volume. The capillary is then placed on a glass slide and sealed with two-part epoxy (BSI-202, Bob Smith Industries). The sample is cured for roughly 30 min. Care is taken to ensure that no air bubbles are present in the tube during sealing. Ultimately, these slides are placed into an acrylic holder on the microscope stage that positions the capillary facing the objective of the microscope.

## Brightfield imaging and counting crystals

Brightfield microscope images are obtained using a Nikon Ti2 microscope with a 10×-magnification, 0.45-NA objective (MRD00105, Nikon), a 1.5×-magnification tube lens, and a Pixelink M12 Monochrome camera (M12M-CYL, Pixelink) connected to a desktop computer. The focus is set such that a majority of the presented faces of the crystals are in focus. To maintain focus throughout the duration of the experiment, we use the Nikon Perfect Focus System.

## Polarized light imaging

Polarized reflection and transmission microscopy images were taken either on an Olympus BX51 microscope with an incandescent lamp or a Nikon Ti2 microscope with a white LED illuminator. Crossed polarizers are installed in each case, and the images are taken with a color CMOS Camera (CS895CU, Thorlabs). The analyzer is always aligned perpendicular to the polarizer for maximum contrast. We image some crystals at different angles relative to the polarizer. For these images, the angle of the polarized light is shifted by 5° between each image by rotating both the polarizer and the analyzer in tandem. It is only necessary to image 90° of rotation as the other quadrants are symmetrical. To obtain polarized light images of crystals while they were growing, the sapphire window on the Peltier unit had to be replaced by a glass one, as the sapphire affected the polarization of the incident light.

## Fluorescent confocal imaging

To determine the crystallographic structure, we image the crystals with a Leica SP8 laser-scanning confocal microscope. Since our crystals are composed of two particle species, we independently dyed each particle type, one with Pyrromethane 546 and one with Nile Red. We then take a two-color acquisition to capture the particle locations and compositional order of a given crystallographic plane.

## Seeded growth experiments

Droplets filled with crystals are mixed in an Eppendorf tube with droplets filled with particles that have half the DNA density as compared to the seed particles. To get an acceptably low density of seeds in the final experiment, 1 µL of droplets with seed crystals is added to 4 µL of droplets with weak particles. Then 1 µL of this mixture is added to 4 µL of droplets and so on for a total of three dilutions. Finally, a fourth dilution adds 1 µL of this diluted mixture to 9 µL of droplets containing weak particles. This solution is then loaded into a capillary until the capillary is completely full, sealed using UV glue (Norland Products, NOA68), and then cured for at least 10 min under a mercury vapor UV lamp.

The sample is brought to a microscope and is heated to a temperature at which the weak particles disassociate, but the seeds remain intact. The sample is then quickly brought to an analytical balance with an attached ionizer (Mettler Toledo XSE104) and is gently moved back and forth across the ionizer aperture for 30 s. The ionizer breaks the emulsion, combining the particles with the seeds. The sample is put on the microscope again and is heated using a thermoelectric cooler to melt the weak particles until the combined system is in equilibrium[19].

The field of view is centered on a region with the desired density of seeds, and a time-lapse video of growth is recorded. We acquire one picture every five minutes. To maintain focus over the duration of the experiment, we use the Nikon Perfect Focus System. On each image, the projected area of a chosen reference crystal is measured using image analysis routines written in Matlab. The system starts at a temperature at which crystal growth does not occur, and the temperature is automatically lowered in 0.05 °C steps until the reference crystal begins to grow.

## Reporting summary

Further information on research design is available in the Nature Portfolio Reporting Summary linked to this article.

## Data availability

The data that support the findings of this study are available from the corresponding author upon request.

## Code availability

All source codes central to the conclusions of the current study are available from the corresponding authors upon request.

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

## Acknowledgements

We thank Larry Luster for the initial conversations about seeded growth. This work was supported by the National Science Foundation. A.H., H.S., and W.B.R. acknowledge funding from DMR-1710112 and DMR-2214590; T.E.V. acknowledges funding from the Brandeis Bioinspired MRSEC (DMR-2011846).

## Author contributions

A.H., W.M.J., and W.B.R. conceived the project. A.H. performed and analyzed the microfluidics-based crystallization experiments. A.H. and H.S. performed and analyzed the seeded-growth experiments. T.E.V. and H.S. characterized the crystal structures. T.E.V. analyzed the crystal structures and developed the Winterbottom reduction model. W.M.J. developed the theoretical models. A.H., T.E.V., W.M.J., and W.B.R. wrote the paper with input from all authors.

## Competing interests

The authors declare no competing interests.
