## [Peer Review File · Nature Communications]

Comments for Author

The manuscript by Hensley et al. describes a seeded growth method for the synthesis of macroscopic DNA-programmed photonic crystals. The manuscript should be re-considered for publication after the comments below are addressed:

1. The crystal structure is not properly analyzed.
 - a. First, the authors cannot rule out the possibility that a CsCl crystal structure exists (similar examples can be found in *Nat. Commun.*, 2012, 3, 1209; *Nature*, 2020, 580, 487.), rather than the CuAu structure as claimed. In addition, it is possible that the structural identification in another recent publication by the same authors (*PNAS*, 2022, 119(1), e211405118) could be partially incorrect.
 - i. CsCl crystals are generally reported to develop a characteristic rhombic dodecahedral crystal habit (Wulff construction of the CsCl lattice, as shown in the following model), which is consistent with what the authors observed in the previous *PNAS* paper (Fig. 4D). The Wulff construction of the CuAu lattice should not exhibit this shape [since the (110) facet energy is too high].

- ii. The CsCl crystals have a better-matching (100) view, as was reported in the previous *PNAS* paper (Fig. 1C)

- b. The structural identification was done by imaging one crystal, but there could be multiple types of crystals that co-exist in a single batch – Fig. 2D and Fig. 4B (green-yellow) show crystals with projection angles of ~ 125 degrees, which is consistent with the projected angle of a rhombic dodecahedron. However, Fig. 4A, 4B (red) shows projection angles of ~ 120 degrees, consistent with that of a hexagonal prism (a crystal habit often developed by CuAu lattices).
 - c. Therefore, it is possible that both CsCl and CuAu crystals could exist in one batch, which is consistent with the report from Nat. Commun., 2012, 3, 1209.
 - d. More careful structural identification should be done using 3D reconstruction under confocal microscopy (Nature 2020, 585, 524) or small angle X-ray scattering. These necessary additional characterization data will also reveal whether these are actually “single crystals” as the authors claim (not fully supported currently).
2. If the crystal structures were indeed misidentified, then the authors should conduct a more careful *in situ* or statistical study (ideally both) of whether a structural evolution or symmetry change can happen during seeded growth?
3. The novelty of this report is reduced owing to the recent PNAS paper from the same authors. The same growth procedure was used in this manuscript, except for the seeded growth portion. What new science is presented in this manuscript? Please explain and emphasize.
 - a. If the authors aim to demonstrate a new and better method for making large-scale single crystals (granted they are CuAu crystals), they would be isostructural to the fcc opal crystals that were already reported and that could be achieved on the centimeter scale years ago – a “macroscopic photonic crystal” that is structurally no different than what was achieved in this work.

- b. If the authors aim to report that the “seed-mediated” crystal growth is new, I would argue that there are similar examples that can be cited using DNA-mediated crystal engineering (ACS Nano 2016, 10, 1771).
 - c. If the authors mean to deliver a fundamental investigation of crystal growth, the crystal structure evolution (at least, if it exists) should be fully studied and the structures of each crystal identified.
4. In Fig. 1A, the authors mention “same-size single crystals.” The dimensions of the seeds seem to be varied as shown in the image. How do the authors define “same-size”?
5. The seed-particle melting temperature and the weak-particle melting temperature are roughly equal, so the seeded growth temperature could potentially melt the seeds, or the nucleation from the bulk during the seeded growth may not be totally suppressed. The authors should investigate how the melting temperature differences influence seeded growth.
6. For Fig. 4B, “an overview of the reported sizes of the largest crystals from the literature as a function of the constituent particle size.” The cited papers are not for the “largest” reported sizes. For example, Nature 2021, 591, 586; Nature 2020, 580, 487; and J. Am. Chem. Soc. 2010, 132, 289.
7. In page 4, there are two “are’s” in the sentence: “...we dilute the seeds until they are are separated by at least...” Please edit the manuscript carefully.
8. “We hypothesize that the green-yellow structural color of the 400-nm-diameter-particle crystals arises from second order diffraction.” Please add a citation or simulation to support this hypothesis.
9. Please mention the compositions of the colloidal particles in the main text (instead of only in the supporting information) to give the reader a better sense of the overall structures.
10. In Fig. 2B, “Points represent experiments and lines show the model predictions.” The experimental and the simulated lines are not consistent. What causes the differences between simulation and experiment?

Response to Reviewers

We thank all four reviewers for their constructive and supportive comments. We have revised the manuscript, the figures, and the supporting materials to address their comments and suggestions, including performing a number of new experiments to quantitatively characterize the 3D structure and composition of the crystals that we self-assemble, demonstrate the appearance of structural coloration upon crystal growth, and more clearly show the single-crystalline nature of the crystals that we prepare by seeded growth. The quantitative analysis approaches that we introduce in the revised version of our submission go above and beyond the methods typically used to determine structure and composition of DNA-programmed colloidal crystals, and could be adopted by others in the field to better characterize the products of colloidal self-assembly. We believe that these changes have strengthened the paper scientifically and clarified the central messages communicated by our manuscript. Below we address the referees' comments in detail and describe changes that we have made.

Reviewer #1 (Remarks to the Author):

Hensley and co-authors report a two-step protocol for the self-assembly of single-domain crystals containing millions of optical-scale particles. This strategy involves the initial assembly of monodisperse crystals in monodisperse droplets made by microfluidics, and the diffusion growth via a seeded diffusion-limited growth process. The assembled macroscopic single-domain photonic crystals can exhibit some optical properties, like structural coloration, and thus showing some prospect in photonic devices from DNA-programmed materials. This work is systematic; however, the novelty of this work is not clear to me, since the similar work has been reported by the authors in PNAS. I have some doubts if this strategy can be implemented for metamaterial applications, such as laser resonators, which is necessary to prove the advancement of this method. In addition, the authors need to address the following questions.

Response: We thank the reviewer for carefully assessing our manuscript. We wish to emphasize that the novelty of the present manuscript relative to our previous PNAS publication is the focus on growing macroscopic crystals, which is an essential step towards realizing practical DNA-programmed materials. Achieving this goal has been a long-standing challenge in the field and was not accomplished in our prior publication. To this end, we demonstrate, for the first time, that a two-step protocol yields DNA-programmed crystals that are suitable for photonics applications (Figures 3 and 4). Furthermore, we show that the first step of this protocol, assembling small monodisperse crystals in droplets, in fact *cannot* be scaled to produce macroscopic crystals on its own as a result of fundamental physical reasons, as we explain in the manuscript. This limitation of the droplet technique, which we establish through a combination of systematic experiments and theoretical modeling (Figure 2), has not been discussed in prior literature. Finally, we clarify in the revised manuscript that our two-step protocol is applicable to crystals with different sized colloidal particles and thus different crystal

structures. We believe that these aspects constitute a major scientific and technological advance over prior work.

(1) The property of structural coloration is very common in photonic crystals, which does not show the application value of this work. It is suggested to add the relevant experiment to prove the metamaterial applications of single-domain crystals, such as laser resonators.

Response: We thank the reviewer for their comment. However, we disagree with their assessment that demonstrating structural coloration of DNA-programmed colloidal crystals does not show the application of this work. The central focus of our manuscript is on developing and understanding a rational and robust platform for assembling macroscopic, single-crystalline materials from DNA-coated colloids. Lacking such a rational and robust approach to colloidal crystallization with DNA, the field has yet to demonstrate any programmable bulk material property from optical-scale colloids, including optical metamaterial properties like structural coloration, despite nearly 20 years of research activity.

While the reviewer is correct that the ultimate promise of DNA-programmed crystallization is loftier than structurally colored materials, the development of complex devices, like programmable laser resonators, is a scientific challenge in and of itself, and is therefore beyond the scope of this article. We have revised the introduction and the conclusion of our manuscript to better contextualize our advance.

(2) Does this strategy to prepare large single-domain crystals have any requirements for materials? Is there a limit to the size (maximum or minimum) of the colloidal material?

Response: We have added a comment about the requirements for two-step seeded growth to the discussion. In brief, the only requirement on the particle size is that the particles be Brownian since our method relies on the near-to-equilibrium assembly of the seeds and the diffusion-limited growth of the large crystals. This places a practical upper bound on the particle diameter of roughly 2 micrometers. In principle, there are no constraints on the particle composition as long as the interparticle interactions arise solely from DNA hybridization. Therefore, the field should be able to apply our new platform to control the assembly of the wide range of materials already used to make colloidal crystals from DNA-coated particles, including polymers, metals, metal oxides, semiconductors, and magnetic materials.

The comment is reproduced below for convenience.

“... because our data-driven modeling of nucleation and growth could be extended to other particle sizes and crystal structures, our two-step platform should work for any Brownian suspension of DNA-coated colloids (i.e., particles with diameters less than roughly two micrometers). Similarly, because the interactions that drive nucleation and growth are due entirely to DNA hybridization, our method should also apply to a wide range of particle

compositions, including polymers, metals, metal oxides, semiconductors, and magnetic materials.”

(3) Fig 3-B demonstrates the growth of a single seed over roughly one day. As the seed grows, it shall exhibit the property of structural color, which is also an important aspect to demonstrate its growth. However, the images in Fig 3-B cannot show this feature. It is suggested to optimize them and add the microscopic color images.

Response: We thank the reviewer for this fantastic suggestion! The reason that we do not see coloration upon growth in our original Figure 3B is that the colloidal particles are 600 nm in diameter and therefore the primary and secondary Bragg peaks are expected to be in the infrared region. To test the reviewer’s hypothesis, we perform the same experiment using 250-nm-diameter particles instead (see Fig 3C in the revised manuscript). As anticipated, we clearly see the appearance of coloration as well as a marked increase in the color saturation upon crystal growth. As the reviewer suggests, this experiment shows a direct connection between the bulk optical properties (e.g., coloration in this case) and the crystal size.

(4) What is the maximum size of the single-domain crystals that can be grown using this strategy? What factors can hinder its growth?

Response: In principle, there is no upper limit to the size of crystals that can be assembled using our method provided that the seeds are at a low enough density that their concentration fields do not overlap and that the sample chamber is sufficiently large. In practice, we stop our growth experiments once the crystals reach roughly 100-200 micrometers in linear dimension because reaching this size takes a couple of days of continuous monitoring on our optical microscope.

(5) From the experimental results in the manuscript, the shape of the grown single-domain crystals is a fixed regular hexagon. What is the reason for this? Is it possible to grow single crystals with other shapes?

Response: In the revised manuscript, we have included extensive characterization of the 3D crystal structure, composition, and habit. Because Reviewer 1 focuses on the crystal habit, we will describe the nature of the crystal habit here and revisit the crystal structure and composition below.

In general, the crystal habit (i.e., the characteristic shape of a single-domain crystal) is determined by the underlying crystal symmetry and the detailed nature of the crystallization process itself. In our experiments, the crystal habit appears to be well described by the Wulff construction, modified to account for the fact that the crystals grow near to a surface. The Wulff construction is a general approach for determining the equilibrium shape of a crystal that

minimizes the total surface energy of a crystal of a given volume. Because the various crystallographic planes have different densities and compositions, they also have different surface energies (and surface energy densities). Therefore, the surface facets will spontaneously arrange themselves to obtain the crystal habit of lowest surface energy.

This process is modified in our experiments owing to the fact that, once large enough, the late-stage crystal growth happens near to the bottom coverslip surface. Because growth in the direction of the coverslip surface is slowed, the crystal adopts a truncated Wulff shape as if it were 'sliced' by a plane that is parallel to the coverslip. This modification to the Wulff construction bears some similarities to the Winterbottom construction, therefore we refer to it as the "Winterbottom reduction."

We have included a section in the SI describing the origin of the crystal habit and we reproduce Fig. S10 here for convenience.

Panel A shows an example prediction of the single-crystal shape due to minimization of the total surface energy, as well as the two distinct views of the crystal habit. Panel B shows how those two views would be modified in our experiment due to the presence of the coverslip. The net effect of the coverslip is to alter the crystal facets from having four sides (as in A) to six sides (as in B).

Given the origin of the crystal shapes that we observe, described above, we conclude that it would be possible to grow crystals with different habit geometries by changing the symmetry of the underlying crystal lattice. Furthermore, it may also be possible to change the crystal morphology by conducting crystallization under conditions where the crystal habit is determined by the growth kinetics as opposed to minimization of the total surface energy. Two examples would be the growth of hexagonal prisms [1] or crystalline needles [2], as observed in previous reports.

Reviewer #2 (Remarks to the Author):

The manuscript is a continuation of the recent work for the same group, published in PNAS (Hensley et al. PNAS 2022). In the PNAS paper, the authors mainly studied the homogenous nucleation of DNA-coated colloids, and in this manuscript, they extended their strategy to seeded-growth of colloidal crystals by DNA-coated colloids.

The strategy described in this manuscript has two steps. The first step is to fabricate crystal seeds inside emulsion droplets. To optimize the experimental conditions, the authors varied several experimental parameters, such as particles concentration, droplet volume and temperature ramping procedures. In the second step, the authors supplied crystal seeds with colloidal particles that have a lower DNA grafting density, so that the growth of crystals can occur at a higher temperature where the secondary nucleation is suppressed. They showed that this two-step protocol can lead to colloidal crystals that are 100-um in the x-y plane. Overall, I find this paper is clear and well-structured, and can be considered for Nature communication after a revision.

I have the following questions and comments:

1. Both the authors' PNAS paper and this manuscript worked on AuCu-FCC crystals only. To show the robustness of this approach, it's better for the authors to show that different types of crystals can be self-assembled. Ref. 14, 15, 17, 20, 26 are from journals of the same level, and they all demonstrated more than one type of crystals.

Response: We thank the reviewer for this suggestion and we aim to explore a wider variety of crystal structures going forward. At present, we have demonstrated the robustness of our approach by showing that it works for three different particle sizes that produce three different crystal structures: one that is very close to FCC-CuAu, one that is CsCl, and one that is intermediate between those two structures (a binary, compositionally ordered body-centered tetragonal (BCT) crystal). Two of the papers that the reviewer mentions, Refs 15 and 20, also only demonstrate three different crystal structures each. While it is true that Ref 17, for example, shows four different crystal symmetries, the focus of that article is on expanding the structural diversity of colloidal crystals from DNA-coated colloids; i.e., the paper is focused explicitly on making many different types of crystals. The central focus of our manuscript is on developing and understanding a rational and robust platform for assembling macroscopic, single-crystalline materials from DNA-coated colloids.

2. This manuscript lacks the crystal characterization in the third dimension (z-direction). Because of gravity, seeds can sediment to the bottom very fast, and the crystal growth in the third dimension might be different. The claim that these are single crystal follows from only surface microscopy. It would be important to have a cross section or other method to probe a z-direction.

Response: We thank the reviewer for this fantastic suggestion and we have performed the 3D characterization that they recommend. We find that the crystals are indeed single-crystalline in all three directions and that the crystal structure is uniform throughout. In other words, the local crystal symmetry and composition is the same in every region of the crystal that we can image, including in the z-direction. We have summarized these results in the Supplementary Information. We reproduce SI Figure S8 below, which shows the XY in-plane (110) crystalline order as well as the YZ perpendicular plane (001) crystalline order. Both show clear crystalline order and have the same crystal structure. We note that the image quality of the YZ (001) plane is lower due to light scattering from the polystyrene particles. However, the image quality is still sufficiently high to see 8-9 crystal planes into the crystals. Panel A shows the two perpendicular views of a crystal formed from 600-nm-diameter particles; panel B shows two perpendicular views of a crystal formed from 430-nm-diameter particles. We were unable to image the crystals of 250-nm-diameter particles because the particle size is just at the diffraction limit of our optical microscope.

3. Does the annealing protocol also suppress other crystal defects? For example, point defects and dislocations. From the microscopy image, I see that there are still some point defects.

Response: As the referee correctly points out, both the single crystalline seeds and the large crystals prepared by seeded growth have defects. For example, the crystal in Figure 1A shows a few vacancies and a substitutional impurity (an anomalously large red particle.) Although a dedicated study of the types and densities of defects would be required to answer this question definitively, our preliminary evidence suggests that the most abundant defect in the crystalline seeds are vacancies (there are not many substitutional impurities given that we purify the particles by density gradient centrifugation and there are no grain boundaries). Because our experiments are performed at finite temperature, a certain density of vacancies would be expected even in thermal equilibrium, but we do not know how close we are to this limit.

Our seeded growth experiments do show the incorporation of additional defects upon crystal growth, as seen by the small chips and cracks in the micrographs in Figures 4C-E. These defects could be due to the close proximity of the coverslip surface, where it is possible for particles at the growth front to become pinned to the coverslip surface. Future experiments with different surface passivation schemes would be required to test this hypothesis.

4. In Page 4, the last paragraph, the authors claimed that

“In fact, the crystal shown in Figure 4A is orders of magnitude larger in volume than any crystal made from similar-size particles reported in the literature, highlighting the strength of our method.”

I doubt this. See for example the paper: Hueckel et al., Ionic Solid from Common Colloids, Nature 2020. In this paper, Hueckel et al. used 600-nm PS particles and the colloidal crystals are on the order of 100 micron, similar to the crystal size in this manuscript. They used a combination of electrostatic interaction and steric repulsion. The authors may want to rephrase the statement to “crystals larger than any crystals made from DNA-coated colloids”, as what the authors did later in this manuscript.

Response: We apologize to all four reviewers for this miscommunication. We intended to compare our results against the published literature on crystallization of DNA-coated colloids. There are indeed many beautiful experiments that show assembly of macroscopic crystalline materials from colloids *using other types of colloidal interactions* (e.g., the electrostatic attractions that the reviewer mentions). The omission of this clarifying detail was entirely our fault and we have revised the manuscript accordingly. We now explicitly state that we are comparing our results to articles that focus on crystallization of DNA-coated colloids alone. We have also included a discussion highlighting other methods for making macroscopic materials from colloidal crystallization, while noting the caveat that these other methods do not yet have the same level of programmability as DNA-coated colloids, and therefore, have not yet yielded the same diversity of crystal lattices.

5. For photonic crystal applications, these crystals have to be dried out of a suspension. I wonder if the authors have tried to do this. The challenge is that the suspension contains a lot of salt, and to get a clean crystal, the authors need to rinse the crystal several times and, in this process, the crystals might collapse. (although crystals in this manuscript are FCC-like, they have a higher volume fraction)

If the authors can dry the crystals out, they can also characterize the crystals in the third dimension, which would be very difficult to do by optical and confocal microscopy because of the refractive index mismatch (PS vs Water).

Response: We have not tried to dry these crystals from suspension. Fortunately, because the particles are small enough that their scattering cross section is not too large, we are able to image within the bottom 5-10 layers of the crystal to determine the 3D composition and structure.

Following the reviewer's advice, should we attempt to dry out our crystals, we will attempt to UV-weld [3] the interparticle connections before washing out the salt to prevent the crystals from collapsing. This work is ongoing.

6. The feedback system that is used to continuously monitor crystals by modulating the temperature seems to suggest that not volume loss but instead a chemical potential gradient is being biased via temperature to push the reaction towards crystal growth.

Response: We thank the reviewer for highlighting that we failed to clearly articulate the origin of the feedback loop that monitors crystal growth. We have revised this section of the SI to clarify the physical picture underlying the feedback loop and why we conclude that it is needed to overcome a weakening in the interparticle interactions rather than simply a lowering of the chemical potential gradient.

In brief, we implemented the feedback loop that monitors crystal growth in response to the observation that the growing crystals, and eventually the seeds themselves, will melt if held at a constant temperature. The initial temperature for the secondary growth stage is chosen empirically such that the seeds grow (i.e., so that the solution is supersaturated relative to the equilibrium concentration of colloidal particles in the gas phase) at the start of the protocol. Thus, we know that this temperature is lower than the crystal melting temperature when the secondary growth process is initiated. However, after some time, we observe that crystal growth actually *reverses*, and the crystals begin to decrease in size, when the temperature is held constant. This observation implies that the *melting temperature* of the crystal, and thus the equilibrium concentration of colloidal particles in the gas phase, is changing in time. By contrast, a decrease in the chemical potential gradient, which is inevitable in the long-time limit, would only reduce the rate of crystal growth as the concentration of colloidal particles in the gas phase approaches the equilibrium concentration, and would not be expected to lead to crystal melting.

The only way for the crystals to melt at a fixed temperature and fixed total concentration of colloidal particles is for the binding free energy between the constituent particles to decrease. While the binding free energy could decrease as the result of a few different physical scenarios (e.g., lowering the salt concentration), we hypothesize that it is due to a decrease in the number of accessible DNA molecules grafted onto the particles' surfaces. This hypothesis is supported by two observations: 1) it is known that physical grafting methods like the one we use are sensitive to the presence of surface active molecules, like the free surfactant that is liberated when we rupture the emulsion interfaces; and 2) we do *not* see a similar decrease in the apparent binding free energy in the absence of the surfactant over similarly long time periods (for example, if we perform a nucleation and growth experiment in the very same capillaries

from a solution of particles and buffer that does not come from rupturing a water-in-oil emulsion.)

We have added these details to the discussion in the Supplementary Information.

7. This work is done with one DNA pair. Can they be more descriptive about the relationship between the expected temperature and that found in their work?

Response: We have added a discussion to the Supplementary Information comparing the measured melting temperature to predictions from an experimentally validated mean-field model of colloidal interactions due to DNA hybridization. In brief, we find that the melting temperatures are consistent with the surface densities that are typical for this physical grafting method (roughly 5000 DNA molecules per 600-nm-diameter particle) and the nearest-neighbor predictions of the hybridization free energy from Dinamelt.

8. Can the authors motivate this work with any kind of simulation or modeling to suggest this crystal is “useful”? The trajectory in Figure 4B is not clear. Can the authors comment on the maximum size that is possible using their technique.

Response: This question is difficult to answer in general because what is considered “useful” or not depends entirely on the specific application. For photonic bandgap applications, crystals that are ten or more units cells thick are required to realize the optical properties of photonic bandgaps (as seen in our experiments showing coloration upon growth), but crystals of larger lateral extents (millimeters in size) are more suitable for applications as optical waveguides [4].

In other potential applications, including the use of faceted colloidal crystals as programmable micromirrors for out-of-plane coupling of light, the utility does not necessarily come from the crystal size but rather from the controlled faceting of single crystals [5]. Therefore, it is challenging to define a hard and fast rule regarding usefulness. However, our manuscript does demonstrate our ability to create faceted single crystals with dimensions of a couple hundred micrometers, which are therefore useful for these two specific applications.

As mentioned in the response to Reviewer 2, in principle, our method does not have an upper limit on the crystal size, although the time required for secondary growth becomes extremely long. In practice, we stop our seeded growth experiments after a couple of days.

9. Reference format issue, reference 35

Response: We have removed the URL from the original reference 35.

Reviewer #3 (Remarks to author)

Comments for Author

The manuscript by Hensley *et al.* describes a seeded growth method for the synthesis of macroscopic DNA-programmed photonic crystals. The manuscript should be re-considered for publication after the comments below are addressed:

1. The crystal structure is not properly analyzed:

a. First, the authors cannot rule out the possibility that a CsCl crystal structure exists (similar examples can be found in *Nat. Commun.*, 2012, 3, 1209; *Nature*, 2020, 580, 487.), rather than the CuAu structure as claimed. In addition, it is possible that the structural identification in another recent publication by the same authors (*PNAS*, 2022, 119(1), e211405118) could be partially incorrect.

i. CsCl crystals are generally reported to develop a characteristic rhombic dodecahedral crystal habit (Wulff construction of the CsCl lattice, as shown in the following model), which is consistent with what the authors observed in the previous *PNAS* paper (Fig. 4D). The Wulff construction of the CuAu lattice should not exhibit this shape [since the (110) facet energy is too high].

Rhombic Dodecahedron

ii. The CsCl crystals have a better-matching (100) view, as was reported in the previous *PNAS* paper (Fig. 1C)

Response: We agree with all four reviewers that our structural characterization in the original submission was inadequate. We have therefore performed a new extensive suite of experiments & analyses to answer their questions.

Contrary to Reviewer 3's suggestion, we find definitive evidence that the crystals formed from 600-nm-diameter particles (the same particle size and DNA sequences used in our PNAS article that the reviewer mentions) are not isostructural to CsCl. While it is true that the (100) view of CsCl has a square symmetry, as we often observe (and was shown in Fig 1C of our PNAS paper), it does not have the correct interparticle spacing nor the correct compositional order. As the reviewer illustrates above, the (100) view of CsCl shows a plane of only red (or blue) particles. In contrast, the view that we observe shows a plane with a 1:1 stoichiometry of both red and blue particles (see Fig 1A in the revised manuscript). Furthermore, along the lattice directions in which the particles alternate red-blue-red-blue, we find that the particles are touching, i.e., the center-to-center interparticle spacing is equal to the particle diameter. As seen in the reviewer's illustration, the equivalent distance in the (100) view of CsCl would necessarily be smaller than the particle diameter. Finally, we note that our images are acquired using laser-scanning confocal microscopy, which has a focal depth that is less than a particle diameter. Therefore, we are confident that the two particle species in our images lie in the same plane and not in alternating crystallographic planes as illustrated above.

To directly solve the crystal structures for all three particle sizes we explored, we developed a new quantitative pipeline based on 3D confocal imaging, quantitative image analysis, and crystallography. The three figures below (reproduced from a new section in the SI) show examples for all three sizes from this new pipeline.

In brief, we use laser-scanning confocal microscopy to record 3D image stacks for hundreds of crystals that have sedimented onto a low-index crystal plane (**panel A**). Next we use a particle centroiding routine to find the center positions of each particle, as well as their species identification (i.e., either A (red) or B (blue)). Then we compute the cross-species pair correlation function, $G_{AB}(r)$, which characterizes the likelihood of finding a particle of the opposite species some distance away r . Because of the crystalline order of the assemblies, this pair correlation function exhibits a unique set of peaks whose positions and relative magnitudes depend on the crystal structure: It is like a fingerprint for the crystal type (**panel B**). Finally, we compare our measured pair correlation function to a lookup table of pair correlation functions for low-index planes of various binary crystal structures, including CsCl, FCC-CuAu, and binary body-centered tetragonal (BCT) lattices, to find the closest match (**panel B**). To corroborate this closest match, we also image some of the crystals in a plane orthogonal to the bottom surface and compare the crystal order in the YZ plane to the crystal order in the XY plane (**panels C and D**).

Figure showing the crystal structure of 600-nm-diameter particles.

Figure showing the crystal structure of 430-nm-diameter particles.

Figure showing the crystal structure of 250-nm-diameter particles.

Using this approach, we unambiguously determine the crystal structures for all three particle types. We find that all three crystal structures can be characterized as binary body-centered tetragonal (BCT) crystals, but with unit cells of different aspect ratios, which we characterize through a single parameter C , described in depth in the SI. We highlight that both CsCl and FCC-CuAu also fall under the category of binary BCT lattice structures, with CsCl having $C=0$ and FCC-CuAu having $C=1$.

600-nm-diameter particles. The crystals of 600-nm-diameter particles have a BCT lattice structure with $C=0.7$. As we originally stated, this crystal structure is closest to FCC-CuAu

($C=1$). In fact, the only difference between the $C=0.7$ crystals and FCC-CuAu ($C = 1$), is that the 'like' particles in the $C=0.7$ crystal structure are roughly 30 nm farther apart as compared to FCC-CuAu. This small deviation is nearly imperceptible by eye and therefore would not have been detected by the typical methods that are used to determine the crystal structures of optical-scale DNA-coated colloids.

400-nm-diameter particles. The crystals of 430-nm-diameter particles have a BCT lattice structure with $C=0.2$. To the best of our knowledge this BCT structure has never been reported for a binary mixture of same-sized DNA-coated colloids.

250-nm-diameter particles. The crystals of 250-nm-diameter particles have a BCT lattice structure with roughly $C=0$. Therefore, they are isostructural to CsCl.

From this analysis, we conclude that we do in fact self-assemble three different crystal lattices, one for each particle size, including one crystal lattice structure that has not yet been reported for DNA-programmed crystallization. These findings would not have been possible without the referees' thoughtful comments. Furthermore, in addressing these comments, we have developed a new quantitative pipeline for solving crystal structures from optical-scale colloids that could prove useful to the field at large, since it goes above and beyond the typical approaches used for characterizing colloidal crystals.

b. The structural identification was done by imaging one crystal, but there could be multiple types of crystals that co-exist in a single batch – Fig. 2D and Fig. 4B (green-yellow) show crystals with projection angles of ~ 125 degrees, which is consistent with the projected angle of a rhombic dodecahedron. However, Fig. 4A, 4B (red) shows projection angles of ~ 120 degrees, consistent with that of a hexagonal prism (a crystal habit often developed by CuAu lattices).

Response: We address this comment in our response to comment c) below. In brief, we characterized the structures of hundreds of different crystals and found that, within a single particle size, all of the crystals had the same structure to within experimental uncertainty.

c. Therefore, it is possible that both CsCl and CuAu crystals could exist in one batch, which is consistent with the report from Nat. Commun., 2012, 3, 1209.

Response: After quantitatively characterizing the structures of hundreds of different crystals, we did not find any clear evidence of coexistence between different crystal types in a single experiment. The figure below shows histograms of the "BCT characterization" C for the various particle sizes we explored. Each particle size shows a tightly peaked distribution around a single C value, which decreases upon decreasing particle size. The mean C value for the 600-nm-diameter particles is roughly 0.7, which is much closer to FCC-CuAu than to CsCl. Indeed, we do not find any crystals with $C=0$, indicating that there are no CsCl crystals in coexistence with FCC-CuAu crystals at this particle size. We note that we had originally called

the crystals formed from 600-nm-diameter particles FCC-CuAu because it was the closest reported crystal structure for same-sized binary mixtures of colloidal particles and is indeed exceptionally close to the crystal structure that we observe. As mentioned above, the only difference between the $C=0.7$ crystals and FCC-CuAu ($C = 1$), is that the ‘like’ particles in the crystal structure are roughly 30 nm farther apart in the $C=0.7$ crystal as compared to FCC-CuAu. Furthermore, previous reports on binary colloidal crystallization have used a threshold value to distinguish between CsCl and FCC-CuAu of $C = 0.5$ (Hynnemin, Leunissen, van Blaaderen, Dijkstra, *PRL* 2006). Therefore, using their criterion, we would also conclude that our crystals are FCC-CuAu.

The mean C values for the other two particle sizes are roughly 0.2 and 0 for the 430-nm-diameter and 250-nm-diameter particles, respectively. In other words, the 430-nm-diameter particles assemble into a BCT crystal and the 250-nm-diameter particles form CsCl crystals. At the moment we do not have a theoretical understanding of the crystal structures that we observe. Understanding *why* these crystal structures form will require an exhaustive study itself and will be the focus of a future publication. Importantly, there is no evidence that this variety of crystal structures is the result of our droplet nucleation experiment or our seeded growth experiment since the same crystals form outside of droplets in a simple glass capillary, as well inside of droplets.

d. More careful structural identification should be done using 3D reconstruction under confocal microscopy (Nature 2020, 585, 524) or small angle X-ray scattering. These necessary additional characterization data will also reveal whether these are actually “single crystals” as the authors claim (not fully supported currently).

Response: Please see our response to Question 2 from Review 2.

2. If the crystal structures were indeed misidentified, then the authors should conduct a more careful in situ or statistical study (ideally both) of whether a structural evolution or symmetry change can happen during seeded growth?

Response: Based on the new data presented above, we conclude that the crystal structure of the 600-nm-diameter particles was identified correctly (apart from the slight distortion of the cubic lattice structure, which is on the order of a few tens of nanometers in interparticle spacing). However, we incorrectly assumed that the crystal structures of the small particles (both 250 nm and 430 nm particles) were the same as the crystal structure of the 600 nm particles. As the reviewer requests, this characterization is accompanied by a statistical study of the crystal structures within the ensemble of crystals formed in a single self-assembly experiment. We have updated the manuscript accordingly.

We also do not find any evidence that the crystal structure or symmetry changes during seeded growth. For example, the three different crystals shown in Figure 4c-e exhibit crystal habits that are consistent with our solved crystal structures of crystals of those same particle sizes that are made by slow cooling to room temperature. Therefore, and importantly, we conclude that the particle-size dependence of the crystal structure is not a byproduct of our two-step protocol. However, we do agree with the reviewer that a detailed study of the initial nucleation and growth of the crystals, and whether or not there is any structural evolution during those initial stages, would be quite interesting and will be the focus of our future research in this area.

3. The novelty of this report is reduced owing to the recent PNAS paper from the same authors. The same growth procedure was used in this manuscript, except for the seeded growth portion. What new science is presented in this manuscript? Please explain and emphasize.

Response: We thank the reviewer for this comment and we agree that we should have more clearly articulated the novelty of our work with respect to our recent PNAS paper and other crystallization methods reported in the literature. The new science presented in this manuscript is a quantitative and robust scheme for making macroscopic, single-crystalline materials from DNA-coated colloids. The importance of this work is that it solves a critical and long-standing challenge in DNA-programmed colloidal assembly—making large-scale, faceted crystalline materials—and paves the way to building macroscopic complex crystals that require the full potential of DNA-coated colloids, like the recently demonstrated cubic diamond [4].

We have revised the manuscript throughout to highlight this scientific advance and to better place it within the context of DNA-programmed assembly and colloidal crystallization more broadly.

a. If the authors aim to demonstrate a new and better method for making large-scale single crystals (granted they are CuAu crystals), they would be isostructural to the fcc opal crystals that were already reported and that could be achieved on the centimeter scale years ago – a “macroscopic photonic crystal” that is structurally no different than what was achieved in this work.

Response: The reviewer is indeed correct that other methods for making large-scale crystals from colloids have been reported in the literature. We did not mean to claim that our method is the first to yield macroscopic crystalline materials from colloids. To the best of our knowledge though, our method is the first method to yield macroscopic crystals from micrometer-size DNA-coated colloids. This demonstration is of exceptional importance because the other methods that have been developed for making large scale crystals from colloids have yet to be adapted to assemble large scale crystals with complex crystalline order, like the aforementioned cubic diamond. For example, a common method for making centimeter-scale crystals from colloids is through controlled evaporation of the solvent. This method therefore produces dense packings of colloids, like FCC and RHCP, but is unlikely to yield low density, highly complex crystals like cubic diamond. Indeed, the recently demonstrated assembly of cubic diamond from DNA-coated particles hinges on the equilibrium assembly of complex colloidal building blocks with directional and orientational interactions. It is only through an equilibrium assembly pathway that this crystal structure will form. Therefore, unlike many of the previous processing approaches that yield large-scale crystals that exploit nonequilibrium pathways, our method *uses only near-to-equilibrium assembly pathways to achieve macroscopic crystals*, including both the nucleation and growth steps. While it is true that other equilibrium approaches have been developed recently, see Ref [2] again, extending them to create cubic diamond will require the development of new colloidal building blocks. In contrast, our method could in principle be applied to create macroscopic crystals of cubic diamond from currently available building blocks that assemble via DNA-programmed interactions.

b. If the authors aim to report that the “seed-mediated” crystal growth is new, I would argue that there are similar examples that can be cited using DNA-mediated crystal engineering (ACS Nano 2016, 10, 1771).

Response: We have added a citation to ACS Nano 2016, 10, 1771 as another example of two-step growth of colloidal crystals. However, we highlight a few important differences between our study and this previous report. 1) Whereas this previous report shows that it is possible to grow a thin shell of another colloid composition on top of an existing colloidal crystal, our manuscript demonstrates that a small seed containing only a few thousand particles can be grown by orders of magnitude in size to contain millions of particles while retaining the same crystalline order and composition; 2) Our report introduces a quantitative theory for optimizing the synthesis of the seeds and for identifying the conditions for growth that retain the seeds monodispersity; and 3) Our report also introduces microfluidics as a method for making monodisperse seeds, which is another critical step in making monodisperse crystals via seeded

growth. For these reasons, we believe that our manuscript introduces a number of important advances in crystal engineering beyond those reported in ACS Nano 2016, 10, 1771 or elsewhere in the literature.

c. If the authors mean to deliver a fundamental investigation of crystal growth, the crystal structure evolution (at least, if it exists) should be fully studied and the structures of each crystal identified.

Response: Our report is not focused on the fundamental nature of the structural evolution of crystals upon growth.

4. In Fig. 1A, the authors mention “same-size single crystals.” The dimensions of the seeds seem to be varied as shown in the image. How do the authors define “same-size”?

Response: We define “same-size” crystals as crystals that have a volume that varies by less than 10% from the mean crystal volume. As the reviewer correctly points out, it is true that the projected seed areas vary slightly from crystal to crystal. However, we have quantified the droplet-to-droplet variations in the concentration to be less than 5% of the mean. Because the crystal phase contains all of the colloidal particles at the end of the temperature ramp (i.e., the gas density 30 degrees below the melting temperature is essentially zero), all of the droplets have the same number of particles. Therefore, the variation in the crystal volume for the single crystals must also be less than 5%. Any variation in the projected area that is larger than this variation is simply due to the fact that we are observing different crystal orientations.

5. The seed-particle melting temperature and the weak-particle melting temperature are roughly equal, so the seeded growth temperature could potentially melt the seeds, or the nucleation from the bulk during the seeded growth may not be totally suppressed. The authors should investigate how the melting temperature differences influence seeded growth.

Response: In our experiments, the weak-particle melting temperature is roughly 2 degrees Celsius below the seed-particle melting temperature. While 2 degrees might seem like a small difference, it is in fact a huge difference when compared to the range of temperatures over which single crystals nucleate (roughly 0.1C) or the range of temperatures over which the gas phase has a finite density (roughly 1C). In other words, at the growth temperature, the equilibrium density of the seed-particle gas phase would be roughly 0 and there would be no free seed particles in the solution. This expectation is corroborated by the fact that only the seeds grow and we do not see spurious nucleation that could be caused by free seed particles. Furthermore, if such an issue would arise, we could simply reduce the growth temperature of the weak particles by reducing their DNA grafting density further. Because this issue never arose in our experiments, we did not explore lower DNA densities.

6. For Fig. 4B, “an overview of the reported sizes of the largest crystals from the literature as a function of the constituent particle size.” The cited papers are not for the “largest” reported sizes. For example, *Nature* 2021, 591, 586; *Nature* 2020, 580, 487; and *J. Am. Chem. Soc.* 2010, 132, 289.

Response: We apologize for the confusion, as we only intended to compare crystals formed from DNA-coated particles. We have revised the main text and figure captions to make this point as clearly as possible. We have also added a citation to *Nature* 2021, 591, 586 in our conclusions section, since the method presented therein could be combined with our platform to order materials at even larger length scales.

7. In page 4, there are two “are’s” in the sentence: “...we dilute the seeds until they are are separated by at least...” Please edit the manuscript carefully.

Response: Thank you. We have corrected this typo.

8. “We hypothesize that the green-yellow structural color of the 400-nm-diameter-particle crystals arises from second order diffraction.” Please add a citation or simulation to support this hypothesis.

Response: We have included a new section in the SI that describes a simple estimate of the first and second order diffraction wavelengths. In brief, if the first order diffraction wavelength of the 250-nm-diameter particles is 650 nm (i.e., in the red, as observed), then we would expect a second order diffraction wavelength of the 430-nm-diameter particles to be roughly 550 nm (i.e., in the green, as observed). This calculation is entirely consistent with our hypothesis reported in the main text.

9. Please mention the compositions of the colloidal particles in the main text (instead of only in the supporting information) to give the reader a better sense of the overall structures.

Response: We have added that the particles are made from polystyrene into the main text.

10. In Fig. 2B, “Points represent experiments and lines show the model predictions.” The experimental and the simulated lines are not consistent. What causes the differences between simulation and experiment?

Response: While we agree with the reviewer that the agreement between the predictions and measured single-crystal fractions is not perfect under all conditions, we emphasize that the predictions and measurements are remarkably consistent for a theory with zero fitting

parameters. We further emphasize that this accuracy is surprising given the extreme sensitivity of the nucleation rates in our system to variations in temperature, along with the general challenges of predicting nucleation-dependent phenomena quantitatively from first principles (see, e.g., Reference [6]). For all but the slowest ramp rate ($|dT/dt| > 0.0125$ degrees C/hr) and droplet volumes greater than 0.08 ul, the predicted curves pass through nearly all the experimental data points. The discrepancies at the slowest ramp rate and for droplet volumes smaller than 0.1 ul most likely arise from evaporation, which is most pronounced for longer-duration experiments and droplets with higher surface-to-volume ratios. This hypothesis is consistent with the systematic reduction in the measured single-crystal fractions under these conditions relative to the theoretical predictions, since evaporation tends to increase the supersaturation systematically as the experiment progresses, increasing the probability of secondary nucleation events. However, we agree with the reviewer that these points were not clearly communicated in our previous manuscript. To address this issue, we have reorganized the text as follows:

“Across nearly the full range of parameter space, we find that our model captures our measured single-crystal fractions quantitatively, within the uncertainty of our experimental measurements (Fig. 2c-d). We note that systematic deviations between the predictions and measured single-crystal fraction are observed at the slowest ramp rates, for which the measurements exhibit nonmonotonic dependencies on the ramp rates and droplet volumes. However, we attribute these effects to substantial evaporation of the solvent from the droplets, which can reduce the smallest droplet volumes by as much as 30% during annealing. Therefore, the overall accuracy of our model allows us to predict the conditions required to achieve a target yield and to optimize the temperature-ramp protocol subject to a maximum duration of the experiment.”

Reviewer #4 (Remarks to the Author):

The work by Hensley et al. reports the formation of self-assembled single domain optical-scale crystals. The formation process contains two stages: the monodisperse crystals are firstly assembled in microfluidics-made monodisperse droplets, and then these crystals grow to macroscopic dimensions via a seeded diffusion-limited growth process. The results are interesting and would contribute to the development of photonic devices based on DNA-programmed materials. However, several issues should be addressed before it could be considered for publication in Nature Communications.

1. The advantage of DNA-programmed self-assembly compared with other self-assembly methods should be highlighted, as large-area colloidal crystals can be easily fabricated by other methods (Nat. Mater., 2021, 20, 1512–1518, Chem. Soc. Rev., 2021, 50, 5898-5951).

Response: We agree with the reviewer that we did not clearly articulate the importance of having a platform for making large-scale colloidal crystals from DNA-coated colloids specifically. We have revised the discussion of our manuscript to make this point clearer. As mentioned in our response to Reviewer 3, the central advantage of DNA-programmed self-assembly is that it is highly programmable and can be used to create colloidal crystals with complex symmetries and compositional order, including the aforementioned cubic diamond structure. The other important advantage of our approach as compared to the approach presented in Nat. Mater., 2021, 20, 1512-1518 is that our approach is based on equilibrium assembly, which allows us to design and assemble complex colloidal crystals (e.g., crystals with very low volume fraction like cubic diamond) via minimization of free energy. In contrast, the method presented in the referenced Nat. Mater. article is a non-equilibrium method based on solvent evaporation, which cannot be directly extended to the assembly of these types of complex crystal lattices. Even still, we have added a citation to Nat. Mater., 2021, 20, 1512-1518 and highlighted it as one of the many impressive strategies for making ordered assemblies from colloids and as a point of comparison for our platform.

2. The authors claim that previous work about DNA-programmed assembly was limited by temperature fluctuations, and large-sized single crystals are challenging. In their work, the authors used a precise temperature control system with 0.1°C in step. Therefore, the real challenge is the temperature control or the crystallization kinetics?

Response: We thank the reviewer for raising this important question, which we have now addressed directly in the Introduction. The answer is actually both. First, the extreme temperature sensitivity of the nucleation kinetics of micrometer-sized DNA-coated colloids implies that precise temperature control is required to nucleate a single crystal in a small volume (e.g., in a droplet); even greater precision would be required to nucleate and grow a single crystal in bulk solution. Second, a propensity for kinetic trapping, arising from an effective friction that resists the sliding or rolling of bound particles, is a fundamental feature of

DNA-coated colloids that renders the annealing of polycrystals nearly impossible. We have clarified these important points in the Introduction:

“Assembling macroscopic materials from DNA-functionalized, micrometer-sized colloids is challenging due to the vastly different length scales between the DNA molecules and the colloidal particles (Fig. 1a). This combination leads to crystallization kinetics that are extremely sensitive to temperature and prone to kinetic trapping. The resulting challenges are both practical and fundamental in nature. For example, recent work has shown that crystal nucleation rates can vary by orders of magnitude over a temperature range of only 0.25 °C. Extremely precise temperature control would therefore be required to self-assemble single-domain crystals from a bulk solution (Fig. 1b). At the same time, annealing polycrystalline materials is difficult due to the combination of the short-range attraction and the friction arising from the DNA-mediated colloidal interactions, which slows the rolling and sliding of colloidal particles at crystalline interfaces. Thus, the impracticality of annealing crystals composed of DNA-functionalized, micrometer-sized particles results from intrinsic features of these materials that cannot easily be designed around.”

3. *In the proposed two-step protocol, the formation of a single crystal seed in the droplet plays a critical role. How to certify that the particles in the droplet are assembled into a single crystal? Is there any influence of the unassembled particles on the second growth stage?*

Response: The crystal phase contains all of the colloidal particles within a droplet at the end of the temperature ramp, since the gas-phase density 30 degrees below the melting temperature is essentially zero. Thus, there is no appreciable effect of unassembled particles. (Please see our response to question 6 of Reviewer #2 for a detailed discussion of how we control the temperature in the secondary growth phase, during which particles could in principle detach from the seed).

4. *The suppression of further nucleation through adjusting Δt and realizing isothermal protocol inside the droplet should be discussed in detail, as it is one of the major merits of this work.*

Response: In response to the reviewer’s helpful suggestion, we have expanded our discussion of the theoretical model in the main text. In particular, we have added a new figure panel (Fig. 2b) that explains how the concentration profile is predicted to evolve under an isothermal protocol, and we have refined our discussion of the optimal temperature-ramp step duration as follows:

“Unexpectedly, our theoretical model predicts that utilizing discrete temperature steps, as opposed to a continuous ramp, is in fact beneficial for maximizing the single-crystal yield at a fixed ramp rate, $|\Delta T / \Delta t|$ (Fig. S14; see SI for details). As long as Δt is longer than τ_g , each discrete step can be considered as an isothermal protocol, which optimally suppresses further

nucleation by holding the nucleation rate density, k_n , constant for the entirety of τ_g . By contrast, a continuous ramp implies that k_n increases continuously following the first nucleation event, increasing the probability of secondary nucleation events. Nonetheless, Δt cannot be made too large, as the correspondingly large temperature steps required to maintain the fixed ramp rate would tend to bias the first nucleation event to lower temperatures, and thus nucleation rates that are faster than $1/\tau_g$. Balancing these competing factors, our model predicts that the single-crystal probability is maximized for temperature steps on the order of $\Delta T = 0.1^\circ\text{C}$. As this is the step size used in our experiments, our modeling suggests that further refinement of the precise functional form of our temperature protocol would yield minimal improvement (Fig. S15). In practice, we therefore only need to tune the step duration, Δt , to achieve a target single-crystal yield using a prescribed droplet volume and particle concentration.”

We refer the reader to the SI text and the cross-referenced SI figures for further discussion, since the detailed calculations are quite technical. Nonetheless, we believe that our revised manuscript makes it possible for the reader to follow our argument qualitatively and to understand the key physical principles at play based on the material presented in the main text alone.

5. More characterizations are needed to evidence the single domain crystal property of the assembled structure as well as the solidity of the results.

Response: We thank all four reviewers for their suggestions regarding our characterization of the crystal structure. We have written a new SI section detailing our quantitative characterization of the crystal structure, the crystal habit, and the statistics of the crystal types in each experiment.

6. Page 4, third paragraph, it is hard to tell the symmetry and orientational order of the structure, considering the blurry SEM images.

Response: In response to this reviewer comment, we have performed a new set of experiments in which we image the seeded crystal with single-particle resolution using confocal fluorescence microscopy. This experiment allows us to show convincingly that the crystal has the same symmetry and compositional order throughout, and that the seeded crystal has the same crystal symmetry and crystal habit as the seed. Furthermore, we visualize the seed within the seeded crystal to confirm that the large-scale crystal did indeed grow from a seed and not from homogeneous nucleation. We believe that these new experiments strengthen the article scientifically and we therefore thank the reviewer for this helpful comment.

References

- [1] Seo, S. E., Girard, M., Olvera de la Cruz, M., Mirkin, C.A., "Non-equilibrium anisotropic colloidal single crystal growth with DNA," *Nature Communications*, 9, 4558 (2018)
- [2] Hueckel, T., Hockey, G. M., Palacci, J., Sacanna, S., "Ionic solids from common colloids," *Nature*, 580, 487-490 (2020)
- [3] Gerling, T., Kube, M., Kick, B., Dietz, H., "Sequence-programmable covalent bonding of designed DNA assemblies," *Science Advances*, 4, eaau1157 (2018)
- [4] He, M., Gales, J. P., Ducrot, E., Gong, Z., Yi, G.-R., Sacanna, S., Pine, D. J., "Colloidal diamond," *Nature*, 585, 524-529 (2020)
- [5] Zornberg, Z., Lewis, D. J., Mertiri, A., Hueckel, T., Carter, D. J. D., Macfarlane, R. J., "Self-Assembling Systems for Optical Out-of-Plane Coupling Devices," *ACS nano* (2023)
- [6] Fillion, L., Hermes, M., Ni, R., Dijkstra, M., "Crystal nucleation of hard spheres using molecular dynamics, umbrella sampling, and forward flux sampling: A comparison of simulation techniques," *The Journal of Chemical Physics*, 133, 244115 (2010).

Reviewer comments

Reviewer #1 (Remarks to the Author):

Hensley et al. have addressed most of comments. The addition of Fig 3C in the revised manuscript clearly demonstrates the appearance of coloration. I am also satisfied with the addition of Fig. S10. It is recommended to be published. On the otherhand, I encourage the authors to expand the range of applications of colloidal single crystals. The relevant experiments are suggested to be tried.

Reviewer #3 (Remarks to the Author):

n the revised manuscript, Hensley et al. have performed additional structural analyses that has significantly strengthened their claims, and the addition of these data has improved the quality of this manuscript. However, I have additional concerns about the novelty of this work, which should be addressed before this manuscript is published.

1. The authors recently reported similar work in PNAS (Proc Natl Acad Sci USA. 119, e2114050118 (2022)). Moreover, others have reported in Nature (Nature 610, 674–679 (2022)) on large crystals (>100 μm in size) prepared using DNA-coated nanoparticles. As such, the authors are not the first to show that macroscopic crystals can be produced from DNA-coated nanoparticles. Therefore, the authors should compare their work to that in the previous reports and delineate the features that makes this work novel.

2. Please add a sentence that clarifies how Fig. 4b provides an overview of the reported sizes of the largest crystals of DNA-coated particles from the literature and cite the literature referenced appropriately.

3. Please provide evidence or data to support this statement: "Our two-step protocol breaks this trend, allowing well-faceted crystals to be grown multiple orders of magnitude larger than before."

4. For figure 4b, provide a hypothesis on why these crystal structures have varying particle sizes.

Reviewer #4 (Remarks to the Author):

The authors have fully addressed the issues raised by the reviewer.

REVIEWER COMMENTS

Reviewer 1

Hensley et al. have addressed most of comments. The addition of Fig 3C in the revised manuscript clearly demonstrates the appearance of coloration. I am also satisfied with the addition of Fig. S10. It is recommended to be published. On the otherhand, I encourage the authors to expand the range of applications of colloidal single crystals. The relevant experiments are suggested to be tried.

Response: We thank the reviewer for their recommendation to publish our article as is. We will pursue other applications of colloidal single crystals in future publications.

Reviewer 3

In the revised manuscript, Hensley et al. have performed additional structural analyses that has significantly strengthened their claims, and the addition of these data has improved the quality of this manuscript. However, I have additional concerns about the novelty of this work, which should be addressed before this manuscript is published.

1. The authors recently reported similar work in PNAS (Proc Natl Acad Sci USA. 119, e2114050118 (2022)). Moreover, others have reported in Nature (Nature 610, 674–679 (2022)) on large crystals (>100 μm in size) prepared using DNA-coated nanoparticles. As such, the authors are not the first to show that macroscopic crystals can be produced from DNA-coated nanoparticles. Therefore, the authors should compare their work to that in the previous reports and delineate the features that makes this work novel.

Response: We wish to emphasize that the novelty of the present manuscript relative to our previous PNAS publication is the focus on growing macroscopic crystals from DNA-coated colloidal particles, which is an essential step towards realizing practical DNA-programmed materials with new optical properties in the visible spectrum. Achieving this goal has been a long-standing challenge in the area of DNA-programmed crystallization of micrometer-scale colloidal particles and was not accomplished in our prior publication. On the contrary, in our current submission we show that the first step of our two-step protocol, assembling small monodisperse crystals in droplets, in fact cannot be scaled to produce macroscopic crystals on its own as a result of fundamental physical reasons, as we explain in this manuscript. This limitation of the droplet technique, which we establish through a combination of systematic experiments and theoretical modeling (Figure 2), has not been discussed in prior literature. We believe that these aspects constitute a major scientific and technological advance over prior work.

We have included a reference to the recent Nature article (Nature 610, 674–679 (2022)), which appeared after our initial submission, and updated Figure 4b to include this data point. We

stress, however, that the results in the recent Nature article do not change our conclusions or challenge the novelty of our work because the crystals reported therein are assembled from nanoparticles that are 5-10 nm in diameter. As we have discussed in our manuscript, well-faceted DNA-programmed crystals containing hundreds of millions of nm-diameter particles have been synthesized for years, since the pioneering work of Mirkin and co-workers in 2014 (Nature 505, 73 (2014)). While Mirkin and others have gone on to make many critically important advances in crystal engineering of DNA-programmed, nanoparticle-based colloidal materials, the assembly of the same types of macroscopic, DNA-programmed crystalline materials from micrometer-scale colloidal particles has remained elusive until now. As stated above, the importance of our work is that it establishes a framework for assembling the same types of macroscopic crystalline materials from micrometer-scale (optical-scale) particles using DNA-programmed interactions. This achievement promises to open the doors to new applications in photonics since the particles are comparable in size to the wavelength of visible and infrared light, and exotic crystal structures can in principle be programmed using DNA-mediated interactions. We have made revisions throughout the article to better emphasize this point.

2. Please add a sentence that clarifies how Fig. 4b provides an overview of the reported sizes of the largest crystals of DNA-coated particles from the literature and cite the literature referenced appropriately.

Response: We thank the reviewer for this suggestion. We have updated Figure 4b to include the recent Nature article referenced above, expanded SI Section II.B to more fully explain our methodology for estimating the largest crystal sizes from the literature, and updated the estimated crystal sizes accordingly. The relevant text is reproduced below for convenience.

“Fig. S4 shows estimates of the crystal sizes collected from a review of the literature since 2011. The values for the points, as well as the references, can be found in Table II. Because we ultimately want to make comparisons between similar experimental systems, we focus only on articles in which: (i) the interactions between particles are mediated by DNA hybridization between grafted strands; (ii) the particles are roughly spherical and have isotropic interactions; (iii) an image of an assembled crystal is shown in the main text; and (iv) the image has either clear faceting, denoting the single crystalline nature of the habit, or shows single particle resolution of a single crystalline domain. For crystals that follow these criteria we measure the longest linear dimension of a single domain. For particle diameters that are below 100 nm, the ssDNA brush can increase the lattice spacing of the crystal by a substantial fraction relative to the particle diameter. Therefore, when possible, the lattice spacing is inferred from either the spacing of particles in the images or from the first peak in any provided small angle X-ray scattering (SAXS) data. We then compute the crystal size as the cube of the longest linear dimension divided by the lattice spacing. Again, since we do not consider the packing fraction of the different crystal types that form, these data denote an upper bound on the size of the crystalline assemblies in the literature.”

3. Please provide evidence or data to support this statement: “Our two-step protocol breaks this trend, allowing well-faceted crystals to be grown multiple orders of magnitude larger than before.”

Response: We have added a clarification to the main text that our approach enables the assembly of well-faceted crystals from micrometer-scale DNA-coated colloidal particles that are orders of magnitude larger than before and modified this specific sentence in the figure caption. This sentence now reads “Our two-step protocol breaks this trend, allowing well-faceted crystals of optical-scale DNA-functionalized particles to be grown multiple orders of magnitude larger than before.”

4. For figure 4b, provide a hypothesis on why these crystal structures have varying particle sizes.

Response: We struggled to interpret this comment. To hopefully address the reviewer’s concern, we interpreted the comment in two different ways and modified the text in response to each interpretation. The two interpretations and their responses are given below.

Interpretation 1: For Figure 4b, provide a hypothesis for *why the largest crystal sizes vary with the particle size*.

Our survey of the literature shows that the sizes of the largest crystals assembled to-date from DNA-coated particles decrease with increasing particle diameter. We hypothesize that this trend results primarily from changes in nucleation behavior that depend on the relative range of the DNA-mediated attraction between particles, which is roughly equal to the ratio of the length of the grafted DNA molecules to the diameter of the particles. Because the length of the DNA molecules grafted to nanoparticles and microparticles are typically comparable (order ten nanometers), the relative range of attraction decreases from about 100% of the particle diameter to 1% of the particle diameter going from 10-nm- to 1000-nm-diameter particles (i.e. roughly the range of diameters shown in Figure 4b).

Previous studies of colloid-polymer mixtures have shown that the range of the interaction potential is critical in determining the topology of the phase diagram and the nature of the crystallization process [1]. For example, while such systems show gas-liquid-crystal coexistence for sufficiently long-range attractions, shortening the range of attraction causes the fluid-fluid transition to become metastable [2]. These changes to the phase behavior are accompanied by significant changes in nucleation. For example, depending on the presence of a stable liquid phase or the proximity of a metastable fluid-fluid critical point, it is possible for crystallization to proceed via a two-step pathway, in which a fluid droplet forms first and then the crystal nucleates from within that droplet. The high density of the droplet is predicted to increase the crystal nucleation rate by lowering the surface tension. The high density of the fluid layer surrounding a crystallite also modifies the crystal growth rate by increasing the local

concentration. As the relative range of attraction decreases, the accessibility of this two-step pathway diminishes. While there is no direct evidence that similar changes to the phase behavior and phase transitions occur in DNA-coated colloids, we note that the observed trend of decreasing crystal size (and increasing difficulty in controlling nucleation) with increasing particle size is in line with our expectations based on the colloid-polymer literature. We further note that decreasing the relative range of attraction is also likely to suppress surface diffusion, which tends to slow the growth of small crystallites and may hamper the annealing of multi-domain crystals.

We have modified the discussion of Figure 4b in the SI text accordingly.

Interpretation 2: For Figure 4 c,d,e, provide a hypothesis for why the crystal structure varies with the particle size.

In our previous revision, we definitively showed that the crystals that we produced by our two-step method are indeed single crystalline and have crystal structures that depend on the particle diameter. Figure 4c shows 600-nm-diameter particles that assemble a binary body-centered tetragonal (BCT) lattice that is nearly isostructural to CuAu. Figure 4d shows 400-nm-diameter particles that assemble a binary body-centered tetragonal lattice that is intermediate between CuAu and CsCl. Figure 4e shows 250-nm-diameter particles that assemble a crystal lattice that isostructural to CsCl.

We hypothesize that these crystal structures form due to a balance of specific attraction between 'unlike' particles due to DNA hybridization and nonspecific attraction between both 'unlike' and 'like' particles due to van der Waals interactions. A recent report from David Pine and co-workers shows that polystyrene colloids coated with DNA in the same manner that we use in this article exhibit a weak, nonspecific attraction with an interaction strength of roughly 1-2 kT [3]. This weak interparticle attraction could stabilize the like-like contacts that are present in CuAu and absent in CsCl. They also show that the range of this nonspecific attraction extends beyond the DNA-mediated potential minimum, which could allow the 'like' particles to sit at a slightly greater distance from one another as compared to the unlike particles, again as we see in our BCT crystal structures. Furthermore, theoretical models of the van der Waals interaction predict that the strength of attraction is proportional to the particle diameter [3]. Therefore, we would expect the nonspecific attraction to be weakest for the smallest particle sizes that we explore, which is consistent with our observation that the smallest particles form CsCl (which has no like-like particle contacts).

We have modified the discussion of Figure 4c,d,e and the SI accordingly.

Reviewer 4

The authors have fully addressed the issues raised by the reviewer.

Response: We are pleased to read that we addressed all issues raised.

References

- [1] Whitlam S, Jack RL (2015) The statistical mechanics of dynamic pathways to self-assembly. *Annual Review of Physical Chemistry* 66:143–163.
- [2] Anderson VJ, Lekkerkerker HN (2002) Insights into phase transition kinetics from colloid science. *Nature* 416(6883):811–815.
- [3] Cui F, Marbach S, Zheng JA, Holmes-Cerfon M, Pine DJ (2022) Comprehensive view of microscopic interactions between DNA-coated colloids. *Nature Communications* 13: 2304.

Reviewer comments further

Reviewer #3 (Remarks to the Author):

The authors have addressed all of my questions and I recommend publishing this manuscript in Nature Communications.

REVIEWER COMMENTS

Reviewer #3 (Remarks to the Author):

The authors have addressed all of my questions and I recommend publishing this manuscript in Nature Communications.

Response: We are pleased to read that we have addressed all issues raised and that the Reviewer recommends that our article be published in Nature Communications.